# Focusing: View-Consistent Sparse Voxels for Efficient 3D VAE

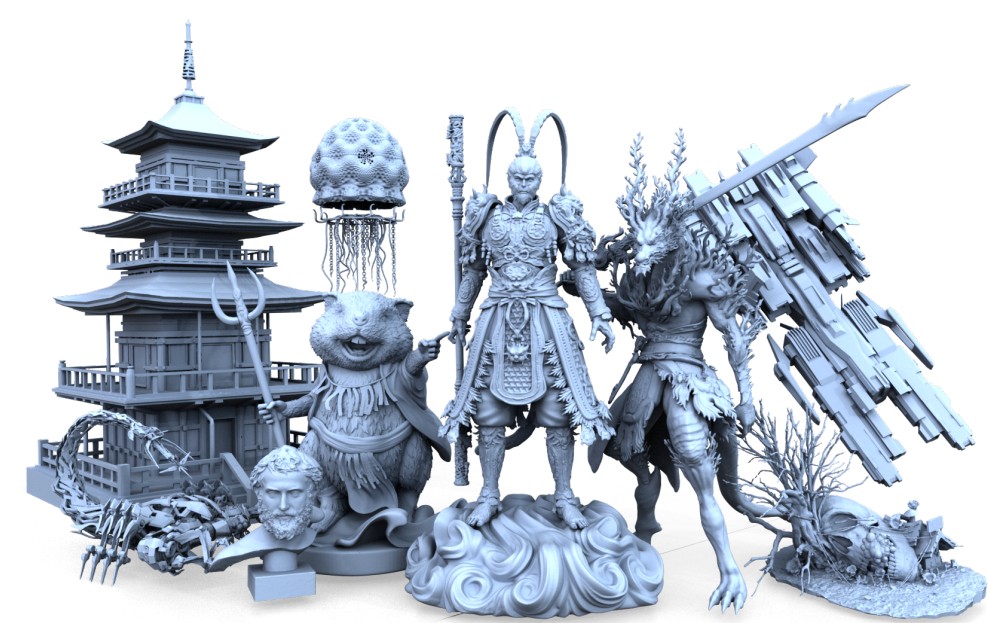

Figure 1: **Focusing** is an efficient and effective 3D VAE, capable of generating high-fidelity 3D models at $1024^3$ resolution using less than 50GB of memory by using the render loss with an efficient voxel selection.

## Abstract

High-fidelity 3D generation remains difficult. Although some methods have proposed converting raw meshes to SDFs, it remains a lossy process. TripoSF presented a VAE training paradigm based on a rendering loss to circumvent this lossy SDF conversion, achieving high-precision surface reconstruction. However, because the rendering loss cannot supervise all the VAE outputs in the same way as SDF supervision, it limits detail and scalability. We present **Focusing**, a 3D VAE that improves efficiency by activating only the voxels that matter for a given view. Our key idea is a depth-driven voxel carving performed in the structured latent space: voxels inconsistent with the rendered depth are pruned before decoding. This concentrates learning on locally relevant geometry, reduces attention and decoding costs, and lowers video random access memory (VRAM) usage. To stabilize training and capture fine details, we further introduce an adaptive zooming strategy that adjusts camera intrinsics to keep the number of active voxels within a target range. The VAE is trained with a render-based loss on depth, normals, masks, and perceptual terms, and we add simple regularizers (e.g., sparse-voxel TV and a short warm-up with TSDF supervision) to reduce small holes and speed up convergence. Across standard reconstruction benchmarks, Focusing improves geometric accuracy (CD, F-score) over strong baselines while cutting VRAM consumption, which allows for training the $1024^3$ resolution VAE on as little as 50GB

of VRAM. These results show that local, view-consistent sparsity is an effective route to higher-resolution, more efficient 3D VAEs.

# 1 INTRODUCTION

3D AI-generated content (AIGC) is an emerging research direction with broad applications in embodied AI, digital content creation, gaming, 3D modeling, and AR/XR. Compared to its 2D counterpart, 3D AIGC is considerably more challenging due to the higher spatial dimensionality and the need to model complex geometry and topology. These challenges make achieving high-fidelity 3D generation difficult in a manner that is both accurate and efficient, and the problem remains largely unsolved despite recent progress.

Following the success of 2D AIGC pipelines, most 3D AIGC approaches adopt a two-stage framework. In the first stage, a 3D variational auto-encoder (VAE) encodes a shape representation, such as a signed distance field (SDF), point cloud, or voxel grid, into a compact latent space. In the second stage, a generative model, often a latent diffusion model, operates in this reduced space to synthesize novel 3D assets. Within this pipeline, the 3D VAE plays a central role: its ability to faithfully compress and reconstruct complex geometry directly determines the fidelity of the final results. Improving the efficiency and accuracy of 3D VAEs is therefore a critical step toward high-quality, scalable 3D AIGC.

Since there is no single unified representation for 3D data, recent works have explored different designs for both the input/output and latent spaces of 3D VAEs. Methods such as VecSet Zhang et al. (2023) encode the input into a long tensor representation, but this introduces redundancy, since each latent dimension is correlated with all dimensions of the input. TRELLIS Xiang et al. (2025) addresses this by introducing Structured Latents, where the latent space is represented explicitly as sparse voxels. This structured design improves locality, geometric accuracy, and allows local editing by replacing specific voxels.

Sparc3D Li et al. (2025) and Direct3D-S2 Wu et al. (2025) extend this line by using SDFs for both input and output, thereby enforcing a unified modality. However, since most raw meshes are not watertight, these methods require lossy preprocessing steps to obtain SDFs. To overcome this limitation, rendering-based supervision is adopted by TripoSF He et al. (2025), where the VAE is trained to match rendered depth and normal maps instead of precomputed SDF values. This avoids watertight conversion altogether, making the pipeline more flexible for open or complex training data. To further reduce computation, they adopt Frustum-aware Sectional Voxel Training, which prunes voxels outside the rendering frustum, thereby lowering training costs and enabling $1024^3$ upsampling. While such strategies reduce redundant computation, they also raise a fundamental question: *What is the minimum amount of computation required to render an image from a given viewpoint?*

TripoSF uses a point cloud as input and produces a mesh as output. This setup resembles the classical point cloud reconstruction pipeline, where high-quality meshes can be obtained from dense, oriented point clouds even without neural networks. Unlike TRELLIS, which requires global consistency to reproduce texture, a geometry-focused 3D VAE only needs local point distributions to recover surface patches.

Motivated by this observation and the redundancy of existing approaches, we propose **Focusing**, a local 3D VAE training scheme built on simple yet effective voxel carving. Following the TripoSF framework, we supervise the VAE with depth and normal maps rendered from ground-truth views. Crucially, before decoding, we perform voxel carving as shown in Figure 2: voxels in the structured latent are compared with the rendered depth map, and only those consistent with the view are retained. This discards most irrelevant voxels, enabling the decoder to focus on fine-grained features while substantially reducing VRAM and unnecessary attention computations.

To further improve detail capture, we introduce an adaptive camera adjustment strategy inspired by zooming. By dynamically adjusting the camera's intrinsic parameters, we maintain the number of activated voxels within a controllable range. This mechanism not only stabilizes VRAM usage across diverse inputs but also allows us to expand the field of view (FoV) to capture surface details more effectively. Together, these strategies enable efficient high-resolution training while preserving geometric fidelity. See Figure 1 for examples of high-resolution 3D models produced by our method.

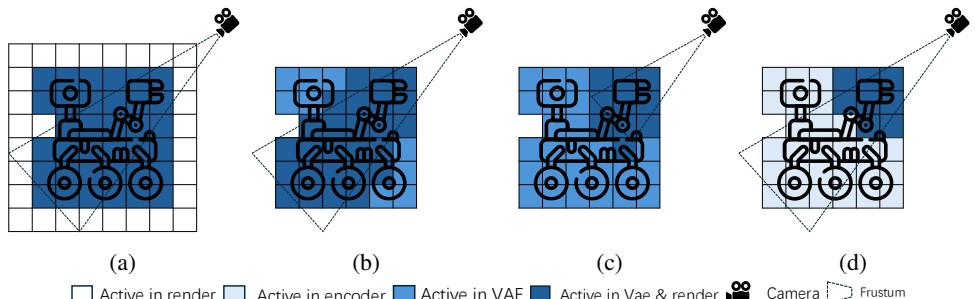

Figure 2: Conceptually illustration of sparse voxels in existing methods and in our approach. (a) TRELLIS encodes 3D models using sparse voxels and generates a mesh on a dense voxel grid for rendering. (b) TripoSF employs SparseFlex to reduce the number of voxels used for mesh extraction during rendering. (c) By adjusting the camera's far plane, TripoSF further reduces voxels that do not contribute to the final render. (d) Our method selects active voxels in the structured latent space based on the depth map. This strategy greatly reduces the computational overhead of both decoding and rendering, and it does not rely on the choice of the far plane.

## 2 RELATED WORKS

### 2.1 3D REPRESENTATIONS

**Meshes.** Meshes are the most common and versatile representation for 3D assets, and they are directly applicable to many downstream applications, such as rendering, animation, and simulation. A line of work explores treating meshes as sequences and generating them with sequence-to-sequence models Nash et al. (2020); Siddiqui et al. (2024); Hao et al. (2024); Gao et al. (2025); Lionar et al. (2025). These methods can produce meshes with stylized or artist-like qualities, particularly quad meshes, but they are typically constrained by sequence length, which limits resolution and geometric fidelity. Other approaches attempt to directly predict mesh topology and geometry from images Wang et al. (2018) or point clouds Hanocka et al. (2020). More recently, differentiable rendering pipelines have been employed to supervise mesh generation with 2D projections. For example, TripoSF He et al. (2025) represents meshes using SparseFlex within a differentiable rendering pipeline, enabling efficient training and supporting higher-resolution mesh generation compared to prior mesh-based methods.

**Point clouds.** Point clouds, often obtained from scanning devices, have long served as input for 3D reconstruction algorithms. Their connectivity-free nature and flexibility make them appealing as neural network input Charles et al. (2017); Qi et al. (2017). Other approaches generate point clouds directly as output distributions Achlioptas et al. (2018); Yang et al. (2019); Liu et al. (2021); Luo & Hu (2021), which allows flexible and high-precision modeling. However, converting point clouds into watertight meshes is both lossy and computationally expensive Kazhdan et al. (2006); Kazhdan & Hoppe (2013); Hou et al. (2022), which limits their use in many downstream tasks.

**Implicit functions.** Implicit functions represent 3D geometry as level sets of continuous fields, such as the zero level set of a signed distance function or the $1/2$-level set of an occupancy field. This formulation is robust and naturally compatible with neural networks, which explains its popularity in both reconstruction and generation tasks Park et al. (2019). Since implicit functions do not directly produce meshes, an additional extraction algorithm (e.g., marching cubes Lorensen & Cline (1987) or its variants) is needed to recover surfaces. However, SDF- and occupancy-based approaches assume watertight geometry, which limits their ability to work with open models. To relax this constraint, UDFs have been proposed Chibane et al. (2020), but existing UDF extraction methods often suffer from inefficiency and instability Zhou et al. (2022); Hou et al. (2023); Ren et al. (2022). A common workaround is to preprocess training data by inflating meshes into watertight versions, yet this conversion can introduce artifacts and reduce geometric fidelity. For example, Sparc3D Li et al. (2025) adopts a flood-filling and deformation-based repair pipeline inevitably leads to losing fine-scale details.

## 2.2 3D VAE ARCHITECTURES

Inspired by the success of 2D diffusion models Rombach et al. (2022), most current 3D generative models follow a two-stage framework: a 3D VAE first compresses the input representation into a compact latent space, and a 3D diffusion model then operates in this reduced space to generate novel shapes. This design allows high-resolution synthesis while keeping diffusion tractable. Unlike 2D AIGC, however, there is no universally accepted compact representation for 3D data. The vast majority of available assets are stored as non-compact meshes, which are not ideal for direct neural network training due to irregular connectivity and variable topology.

3DShape2VecSet Zhang et al. (2023) adopts a VAE to learn a compact vector-set (vecset) representation from point clouds sampled on meshes. The decoder is trained by supervising SDF values at query points, providing implicit geometric supervision during reconstruction. Clay Zhang et al. (2024) extends this approach to large-scale datasets and introduces an inflation-based preprocessing pipeline to enforce watertight meshes. Dora Chen et al. (2025) and Huanyuan2 Team (2025) enhance this framework with importance sampling, which improves the ability to capture sharp features. Hi3DGen Ye et al. (2025) further incorporates normal-map supervision to boost the fidelity of surface detail reconstruction. Despite these advances, vecset representations suffer from significant information redundancy: each feature is correlated with the entire 3D model, making training inefficient and hindering scalability.

Both XCube Ren et al. (2024) and TRELLIS Xiang et al. (2025) replace the global vecset with sparse voxels. In this formulation, each voxel encodes local geometric information, leading to higher-quality reconstructions and more structured latent representations. TRELLIS Xiang et al. (2025) further demonstrates that selectively replacing certain voxels enables flexible local 3D editing. Direct3D-S2 Wu et al. (2025) introduces Spatial Sparse Attention to restrict computations to local neighborhoods, thereby reducing overhead during the diffusion stage. Sparc3D Li et al. (2025) proposes a new preprocessing strategy to improve geometric fidelity. TripoSF He et al. (2025) shifts away from SDF-based supervision and instead employs a rendering loss that compares predicted depth and normal maps with ground truth. This approach avoids the accuracy degradation caused by lossy watertight conversion and allows the VAE to handle both internal structures and open boundaries. However, render-based supervision provides weaker constraints than SDF supervision, since only voxels contributing to visible surfaces are directly trained. As a result, many latent voxels remain under-regularized, which can limit reconstruction accuracy and consistency.

## 3 METHOD

### 3.1 PRELIMINARIES

TripoSF He et al. (2025) introduced SparseFlex, a sparse version of FlexibleCubes Shen et al. (2023), to train 3D AIGC models. SparseFlex is defined by a set of voxels $\mathcal{V}$. Each voxel $v_i \in \mathcal{V}$ contains both its spatial location $(x_i, y_i, z_i)$ (the 3D coordinates of its center) and feature information. Let the number of voxels be $N_v$ and the number of corresponding corners be $N_c$. The feature includes the SDF values $\{s_j \mid 0 \leq j < N_c\}$ and deformations $\{\delta_j \mid 0 \leq j < N_c\}$ for the voxel's eight corners. In practice, the features for each corner are obtained by averaging the values from surrounding relevant voxels. Additionally, the feature also contains the interpolation weights $\{\alpha_i \in \mathbb{R}^8_{>0}, \beta_i \in \mathbb{R}^{12}_{>0} \mid 0 \leq i < N_v\}$ per voxel for Dual Marching Cubes (DMC) Nielson (2004). Formally, the SparseFlex representation, $S$, is defined as:

$$\mathcal{S} = (\mathcal{V}, \mathcal{F}_c, \mathcal{F}_v), \quad \mathcal{F}_c = \{s_j, \delta_j\}, \quad \mathcal{F}_v = \{\alpha_i, \beta_i\}, \tag{1}$$

where $\mathcal{F}_c$ contains the SDF values and deformations at the corner grids, and $\mathcal{F}_v$ contains the interpolation weights for each voxel.

SparseFlex significantly reduces memory consumption and cuts down on unnecessary computational costs. Moreover, by supervising with a rendering loss instead of direct SDF values, TripoSF avoids the lossy watertight mesh conversion process. To further reduce computational overhead, TripoSF introduced Frustum-aware Sectional Voxel Training. This method applies SparseFlex to extract meshes only from voxels that are within the current camera's Normalized Device Coordinates (NDC)

space. By adjusting the camera's intrinsic parameters, this cropping operation also enables the learning of a model's internal structure.

TripoSF then trains a VAE with following losses:

$$\mathcal{L} = \lambda_1 \mathcal{L}_{\text{render}} + \lambda_2 \mathcal{L}_{\text{occ}} + \lambda_3 \mathcal{L}_{\text{KL}} + \lambda_4 \mathcal{L}_{\text{flex}} \tag{2}$$

$\mathcal{L}_{\text{render}}$ is the rendering supervision loss including the following items:

$$\mathcal{L}_{\text{render}} = \lambda_d \mathcal{L}_d + \lambda_n \mathcal{L}_n + \lambda_m \mathcal{L}_m + \lambda_{ss} \mathcal{L}_{ss} + \lambda_{lp} \mathcal{L}_{lp} \tag{3}$$

where $\mathcal{L}_d$, $\mathcal{L}_n$, and $\mathcal{L}_m$ denote the $L_1$ loss for depth maps, normal maps, and mask maps, respectively. $\mathcal{L}_{ss}$ and $\mathcal{L}_{lp}$ denote SSIM loss and LPIPS loss, and are only applied to normal maps. TripoSF culls voxels that are far from the surface during the upsampling process in the decoder. The $\mathcal{L}_{\text{occ}}$ loss is used to guide the self-pruning upsampling module employed by TripoSF to accurately remove these distant voxels. Although this approach can reduce the number of voxels, it also creates more holes and makes convergence more difficult. $\mathcal{L}_{KL}$ is the KL divergence between the learned latent distribution and a standard normal prior, which helps to regularize the latent space. $\mathcal{L}_{flex}$ is the regularization term from Flexicubes that promotes smooth SDF values.

While SparseFlex avoids the accuracy loss from computing SDFs, its training still faces several issues:

- **Irrelevant voxels in the NDC space.** A significant number of invisible voxels participate in mesh extraction and require large VRAM, which has significant influence for sparse voxel resolution scaling up. Although controlling the near and far planes can help, this approach is often suboptimal. To address this, we propose a plug-in-and-play visibility-based voxel carving strategy in Section 3.2 to more effectively reduce the number of active voxels. We also show that this pruning can be performed in the latent space, which greatly reduces the decoder's workload on irrelevant voxels.

- **Rendering blurriness.** Methods based on rendering loss can suffer from blurriness due to the resolution of the rendered image. Capturing accurate detail often requires a higher resolution or a closer camera view, with the former significantly increasing computational cost. In Section 3.3, we introduce an adaptive camera adjustment strategy based on the dolly zoom effect. By controlling the number of voxels to be activated within the view, we can flexibly adjust camera parameters to better supervise the model's fine details.

- **Generating unnecessary holes.** Unlike direct SDF supervision, which provides a strict signal for maintaining watertightness, small holes are difficult to effectively supervise with a rendering loss. In Section 3.4, we introduce our VAE framework and propose a new regularization loss to reduce these holes.

## 3.2 DEPTH-BASED VOXEL CARVING

Our key insight is that the local surface is determined by local points, independent of distant points. As shown in Figure 3, selecting only a subset of voxels in the latent space and feeding them to the decoder can still generate a locally complete mesh, except for some jagged noise at the boundary. Since a rendering loss based VAE does not require a globally complete mesh, we can minimize the number of voxels in the network without affecting the rendering result by aligning the input camera with the filtered voxels.

Specifically, given a camera with extrinsic $\pi$, intrinsics $K$, and the near ($n$) and far ($f$) clipping planes of the viewing frustum, we compute the Model-View-Projection matrix to render the depth map $D$ from the ground truth mesh. For each voxel $z_i$ in the latent code $\mathcal{Z}$, we also project its center to obtain the projected point $z_i^p = \{u_i^p, v_i^p, d_i^p\}$ in the image space and regard $d_i^p$ as the depth of each voxel $z_i$. Using a threshold $r$, we then filter all voxels where $d_i^p > D(u_i^p, v_i^p) - r$ to obtain the view-consistent voxels $\mathcal{Z}_{carve}^p$. To enhance robustness at the image boundary, the $3 \times 3 \times 3$ neighborhood of any remaining voxel is also retained.

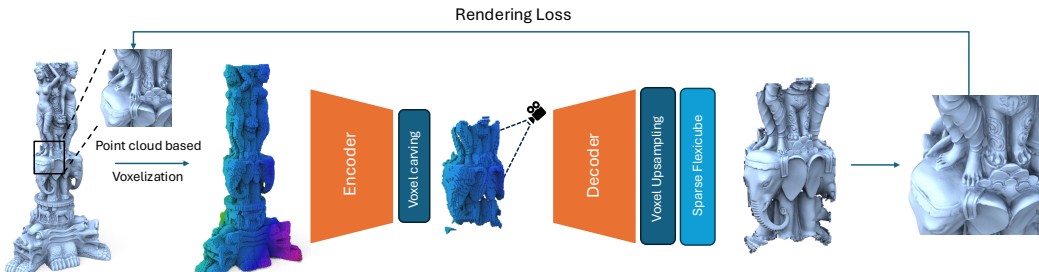

Figure 3: Overview of our framework. Starting from the input mesh, we first voxelize it and aggregate local features from sampled point clouds to form input voxels. A sparse transformer encoder-decoder then compresses these structured features into a latent space, followed by an upsampling module to increase resolution. In the latent space, we perform visibility-based voxel carving to retain only view-consistent voxels that contribute to the visible mesh in the rendered image. The refined structured features are decoded into SparseFlex for final mesh extraction. Supervision is provided by a rendering loss, which compares the rendered images of the input and reconstructed meshes.

### 3.3 ADAPTIVE ZOOMING

The significant variation in the number of visible voxels leads to significant VRAM fluctuations during training. TRELLIS addresses this by removing voxels exceeding a fixed quota, but this can impact the final render if essential voxels are lost. TripoSF utilizes a visibility ratio $\alpha$, to control the number of active voxels, primarily by adjusting the near and far planes. This approach relies on the camera being preset close to the object's surface to capture key details.

In contrast, since we've already removed most invisible voxels, adjusting the far plane has little influence on the number of voxels. We introduce a zooming-based voxel count adjustment method. This approach effectively controls the number of active voxels while enabling flexible zoom-in operations to capture finer model details.

We randomly select a visibility ratio $\alpha \in [\alpha_{\min}, \alpha_{\max}]$ and retain only $\alpha N$ voxels to capture various levels of geometric structure. To limit the maximum number of voxels and avoid having too few, $\alpha N$ is clamped between $N_{\min}$ and $N_{\max}$. We conduct a KNN search in image space to obtain the cropped voxel set $\mathcal{Z}^p_{\text{crop}}$ from $\mathcal{Z}^p_{\text{carve}}$. We then adjust the perspective matrix $P$ using the new image bounding box $(x^n_{min}, x^n_{max}, y^n_{min}, y^n_{max})$ as follows:

$$P_{new} = \begin{pmatrix} \frac{2s}{x^n_{max} - x^n_{min}} & 0 & 0 & -\frac{x^n_{max} + x^n_{min}}{x^n_{max} - x^n_{min}} \\ 0 & \frac{2s}{y^n_{max} - y^n_{min}} & 0 & -\frac{x^n_{max} + x^n_{min}}{x^n_{max} - x^n_{min}} \\ 0 & 0 & 1 & 0 \\ 0 & 0 & 0 & 1 \end{pmatrix} P \qquad (4)$$

where $s$ is the scaling rate used to prevent jagged noise from appearing at the boundary of the image.

### 3.4 VAE STRUCTURE

Following TripoSF, we use a variational autoencoder that compresses the input oriented dense point cloud into a sparse latent code $\mathcal{Z}$ without relying on computationally expensive global attentions. A frozen PointNet Charles et al. (2017) adapted from TripoSF is used to aggregate local geometric features within each voxel. A sparse transformer then utilizes the shifted window attention proposed by TRELLIS to learn relationships between voxels. This process outputs a fused structured latent feature $\mathcal{Z}$, where each voxel possesses local geometric information. The voxel carving and adaptive zooming is then used to filter the voxels in $\mathcal{Z}$ to attain $\mathcal{Z}^p_{crop}$. The decoder takes $\mathcal{Z}^p_{crop}$ as input and uses a series of transformer layers to generate the final output. To support high-resolution output, two self-pruning upsampling modules are then employed to obtain a 2x upsampled result from the initial output, ultimately yielding a 4x resolution output. We only use $\mathcal{L}_{\text{occ}}$ to supervise the self-pruning block and do not use the output of the self-pruning block, $O_{\text{pred}}$, to filter voxels during training, in order to avoid unnecessary holes in the rendered images that negatively impact the rendering loss.

To accelerate VAE convergence and provide initial VAE weights capable of extracting meshes, we supervise $s_j$ using the TSDF $s_j^{raw}$ calculated from raw meshes during the early stages of training:

$$\mathcal{L}_{tsdf} = \sum_{j=1}^{N_c} ||s_j - s_j^{raw}|| \tag{5}$$

To reduce the generation of holes, we introduce the Total Variation loss on sparse voxels to suppress the differences in SDF values between adjacent corners:

$$\mathcal{L}_{tv} = \sum_{d \in [D]} \sqrt{\Delta_x^2(V, d) + \Delta_y^2(V, d) + \Delta_z^2(V, d)} \tag{6}$$

where $\Delta_x^2(V, d)$ denotes the squared difference between the value of $d$-th channel in voxel $v := (i; j; k)$ and the $d$th value in voxel $(i + 1; j; k)$, which can be analogously extended to $\Delta_y^2(V, d)$ and $\Delta_z^2(V, d)$. We apply the TV term above to the SDF grid, denoted by $\mathcal{L}_{tv}$, which encourages a continuous and compact geometry.

The overall loss function is:

$$\mathcal{L} = \lambda_1 \mathcal{L}_{\text{render}} + \lambda_2 \mathcal{L}_{\text{occ}} + \lambda_3 \mathcal{L}_{\text{KL}} + \lambda_4 \mathcal{L}_{\text{flex}} + \lambda_5 \mathcal{L}_{tv} + \lambda_6 \mathcal{L}_{tsdf} \tag{7}$$

## 3.5 RECTIFIED FLOW BASED IMAGE TO 3D GENERATION

Drawing inspiration from TRELLIS Xiang et al. (2025) and TripoSF He et al. (2025), we employ a two-stage rectified-flow generation model, which consists of a sparse structure flow model and a structured latents flow model.

**Sparse structure flow model**. Following the approach of TRELLIS, we first train a Sparse Structure VAE based on voxel carving. This VAE compresses a 3D shape into a smaller resolution using 3D convolutions. Thanks to our preliminary voxel carving process, we can train a high-resolution Sparse Structure VAE with relative ease. Subsequently, we extract DINO features from the condition image and use them as the condition to train a Diffusion Transformer (DiT) via cross-attention. We then employ a rectified flow model to denoise the noised latents.

**Structured latents flow model**. Based on our proposed VAE, we first encode the sampled point cloud and its corresponding sparse structure into the latent space. Similar to the sparse structure flow model, we use the DINO features of the condition image as the condition to train the DiT via cross-attention. We then use a rectified flow model for denoising. Finally, the denoised latents are decoded into a 3D shape by the VAE's decoder.

## 4 EXPERIMENTS

### 4.1 EXPERIMENT SETTINGS

**Implementation Details.** Following TRELLIS Xiang et al. (2025), we train both the VAE and its latent flow model on 183K high-quality assets from Objaverse-XL Deitke et al. (2023). We employ a progressive training scheme for our VAE. The $512^3$ resolution VAE runs on 32 A800 GPUs (batch size 32) with AdamW (initial LR $1 \times 10^{-4}$, weight decay 0.01) for two days. A cosine annealing learning rate schedule with 40K steps is used to adjust the learning rate. We then train the $1024^3$ resolution VAE with 32 A100 GPUs using the same setting.

**Hyperparameter Settings.** We use the same weight configuration as TRELLIS to train our VAE. For our two newly introduced losses, we set $\lambda_5 = 0.001$ and $\lambda_6 = 1$. The $\mathcal{L}_{tsdf}$ is only used for the first 12K steps. We set the threshold $r$ equal to 2/resolution to carve voxels. We use a resolution of $518^2$ to render images and use $s = 1.1$ to calculate the new perspective matrix. For zooming, we set $\alpha_{\min} = 0.1$ and $\alpha_{\max} = 0.3$ to train our model. We set $N_{\min} = 8192$ and $N_{\max} = 15360$ to limit the number of voxels.

## 4.2 VAE Reconstruction Evaluation

Table 1: Comparison of Different Methods

| Method | Input Type | Preprocess | Network Backbone | Texture | Max Resolution |
|--------|-----------|-----------|-----------------|---------|---------------|
| TRELLIS | Visible Only | No | Sparse Voxel | Required | 256 |
| Dora | Watertight Only | Yes | VecSet | No | 256 |
| Direct3D-S2 | Watertight Only | Yes | Sparse Voxel | No | 1024 |
| TripoSF | Inside support | No | Sparse Voxel | No | 1024 |
| Ours | Inside support | No | Sparse Voxel | No | 1024 |

As shown in Table 1, although Dora-VAE, TRELLIS, and Direct3d-S2 all provide VAE weights, they can only reconstruct watertight models. Therefore, it is unfair to directly compare them with our method on raw meshes. We specifically show the comparison with these methods in Section A of the appendix. In the main page, we primarily compare with TripoSF, as currently only TripoSF supports raw mesh reconstruction. Our method demonstrates superior VAE reconstruction performance with quantitative results detailed in Table 2. Our model achieves better metrics than TripoSF at the same resolution in terms of the L2 norm of Chamfer Distance (CD) and F-score with a threshold 0.005. Since even the Level 4 data in the Dora Benchmarks lack sufficient detail, we selected several detail-rich models from online websites to further test our method's ability to capture model details, as shown in Figure 11. Our method demonstrates results comparable to TripoSF's 1024 resolution output. In some high-fidelity models, our 512 resolution result captures geometry details better than TripoSF's 1024 resolution result, as shown in Figure 11.

Table 2: We sample 100K point cloud to measure the Chamfer Distance and F-score on the Dora Benchmark Chen et al. (2025) in different geometry details Levels. Low Chamfer Distance ensures overall shape fidelity, while high F-score ensures local details are accurately covered within an acceptable error radius.

| | Chamfer Distance ($10^{-5}$) | | | | | | | |
|--------|------|------|------|------|------|------|------|------|
| | L1 | | L2 | | L3 | | L4 | |
| | Mean | Std | Mean | Std | Mean | Std | Mean | Std |
| TripoSF$_{512}$ | 1.382 | 0.985 | 1.600 | 1.189 | 2.184 | 1.368 | 3.107 | 2.126 |
| Ours$_{512}$ | 1.353 | 0.995 | 1.513 | 1.008 | 2.116 | 1.351 | 2.806 | 1.771 |
| TripoSF$_{1024}$ | 1.315 | 0.937 | 1.456 | 0.936 | 2.007 | 1.197 | 2.431 | 1.390 |
| Ours$_{1024}$ | 1.294 | 0.886 | 1.429 | 0.873 | 1.901 | 1.011 | 2.264 | 1.055 |
| | F-score | | | | | | | |
| | L1 | | L2 | | L3 | | L4 | |
| | Mean | Std | Mean | Std | Mean | Std | Mean | Std |
| TripoSF$_{512}$ | 0.951 | 0.078 | 0.939 | 0.089 | 0.892 | 0.115 | 0.831 | 0.146 |
| Ours$_{512}$ | 0.953 | 0.080 | 0.945 | 0.086 | 0.897 | 0.117 | 0.841 | 0.147 |
| TripoSF$_{1024}$ | 0.957 | 0.074 | 0.951 | 0.078 | 0.907 | 0.106 | 0.873 | 0.121 |
| Ours$_{1024}$ | 0.958 | 0.070 | 0.953 | 0.073 | 0.916 | 0.091 | 0.886 | 0.097 |

## 4.3 Image-to-3D Generation

We further validate our VAE's utility as a generative foundation model. Visualizations in Figure 7, including image-to-3D results from in-the-wild images, highlight the generalization of our method. The generated 3D shapes maintain sharp edges and rich details while exhibiting high fidelity to the corresponding input images.

Table 3: Comparison of training efficiency with $512^3$ resolution in A100 GPU.

|  | w/o Carving & Zooming | w/o Zooming | Ours ($\alpha_{min} = 0.3$) | Ours ($\alpha_{min} = 0.6$) | Ours ($\alpha_{min} = 0.15$) |
|---|---|---|---|---|---|
| Training Speed (steps/h) | 2652 | 5337 | 8640 | 7590 | 9238 |
| GPU Memory Peak (GB) | 64 | 50 | 32 | 35 | 28 |

## 4.4 ABLATION STUDIES

**Depth-based voxel carving and adaptive zooming.** In this paper, we employ depth-based carving and adaptive zooming to replace the Frustum-aware Sectional method used by TripoSF. This substitution allows for a more reasonable voxel cropping operation. We further find that this cropping operation can be directly applied in the latent space, which further reduces the computational cost. We present the impact of each component on the training speed and memory footprint in Table 3. It demonstrates that performing voxel carving based on visibility in the latent space significantly reduces training cost. Simultaneously, we can further adjust the computational cost by tuning $\alpha_{max}$. However, considering that the encoder still needs to encode global information, an excessively low $\alpha_{max}$ does not provide a linear reduction in cost. Furthermore, because the encoder still encode global information, our method cannot be directly scaled up to train at $1536^3$ resolution. One approach is to increase the upsampling rate from 4 to 8, thereby reducing the computational pressure on the encoder. Additionally, exploring the possibility of performing voxel culling within the encoder would also be an interesting direction.

$r = -2$ / Res $\qquad$ $r = 0$ / Res $\qquad$ $r = 2$ / Res $\qquad$ $r = 4$ / Res

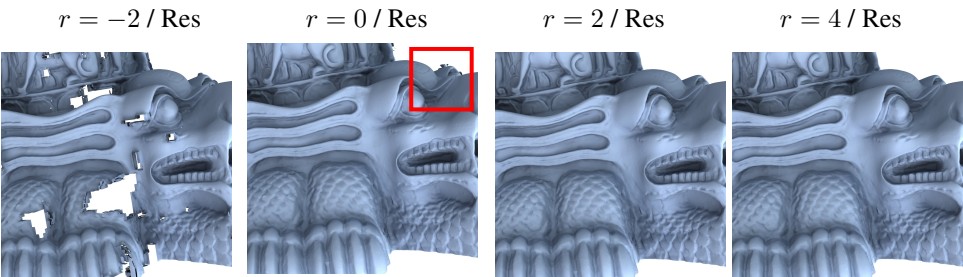

Figure 4: Since the centers of some voxels may lie behind the surface, we use $r = 2$ / resolution as the depth-carving threshold to avoid missing these voxels. A smaller $r$ can lead to geometric errors, while a larger $r$ has almost no effect on geometry but slightly increases the number of retained voxels.

**VAE resolution.** Higher resolution always leads to better VAE reconstruction quality, as shown in Table 2. We further tested the extraction results of the network weights trained at $1024^3$ resolution at different resolutions in Figure 5. We found that our model has the ability to complete $1536^3$ reconstruction without being trained at the corresponding resolution. At the 1536 resolution, chain-beads that were previously not captured were successfully reconstructed.

## 5 CONCLUSIONS & LIMITAION

We present an efficient voxel carving scheme for 3D VAE training and 3D model generation. Only the visible voxels of the structured latent are sent to the VAE decoder for generation. This efficiently removes redundant computation, thereby enabling high-resolution and detailed mesh generation. Experiments demonstrate that our method outperforms SOTA works in terms of efficiency and accuracy.

Currently, our pipeline uses point cloud features as input and is highly dependent on the normal vectors of the point cloud to predict the final result. When the object to be reconstructed is very thin, such as the hair in the last example of Figure 11, it is difficult to obtain a locally consistent normal vector to capture these fine details. In the future, we hope to find a better input feature to overcome this limitation.

| GT | Ours$_{256}$ | Ours$_{512}$ | Ours$_{1024}$ | Ours$_{1536}$ |

Figure 5: We used the $1024^3$ resolution checkpoint to test the reconstruction results at different resolutions. Our method demonstrated generalization capabilities across different resolutions.

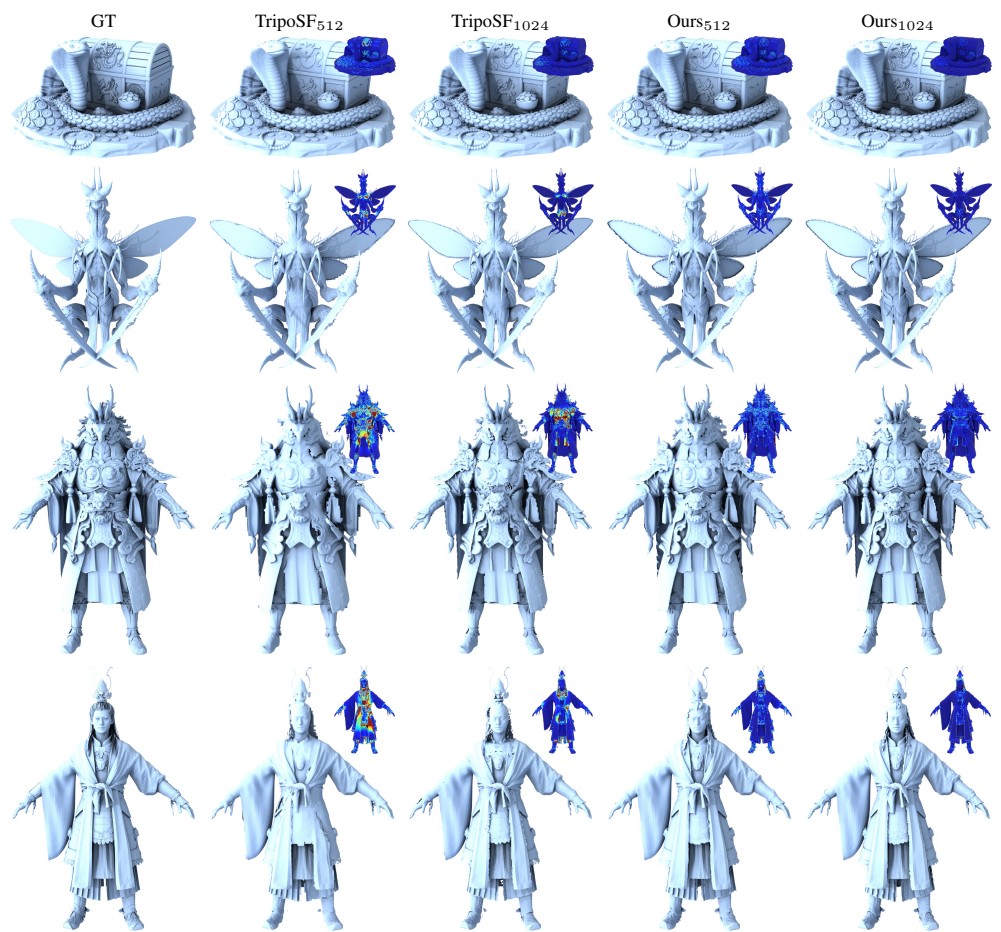

Figure 6: Qualitative comparison of VAE reconstruction between ours and TripoSF with different resolution. Our approach demonstrate superior performance in reconstructing geometry details.

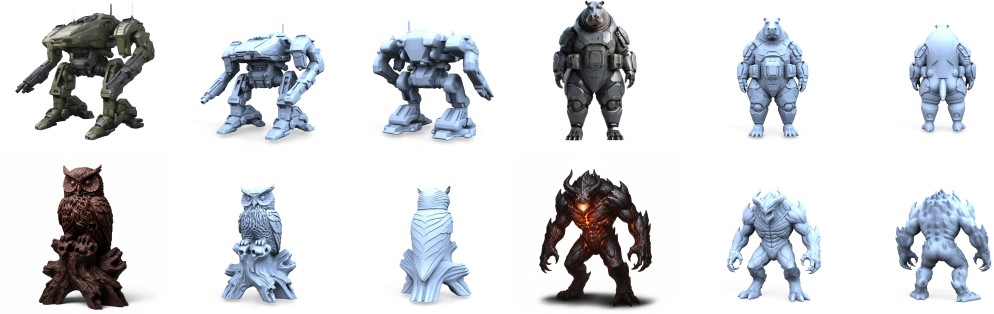

Figure 7: Single image-to-3D generations with in-the-wild images which are collected from Geminis or Dora.

# 6 ETHICS STATEMENT

All datasets and models used in this study are publicly available and have been used in accordance with their respective licenses and terms of use.

# 7 REPRODUCIBILITY STATEMENT

We have clarified our experiment setting in Section 4.1. We will open-source the code and release the trained model.

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

## A    COMPARISION WITH MORE METHODS

We sample 100K points to measure the Chamfer Distance (CD) and F-score on the Dora Benchmark Chen et al. (2025) across different geometry details Levels. Comparison is conducted within each group as shown in Table 4.

**Visible Only.** TRELLIS uses image features as input, and thus can only achieve better reconstruction for the visible regions. Furthermore, since it densifies the network's sparse output using grid points with an SDF of 1. Such operation generates a redundant layer of faces on the inner side of the object, as shown in Figure 8. Although TRELLIS proposed a post-processing method to remove these internal faces, it may lead to incorrect deletions, which could actually worsen the metrics. To avoid misunderstanding and potentially unfair comparisons, we use TRELLIS's visibility-based point cloud sampling results to calculate the CD and F-score.

**Watertight Only.** Sparc3d, Dora, and Direct3d all require the input mesh to be a watertight mesh to compute the SDF. Dora uses the $\epsilon$-level set of the raw mesh as the watertight representation of the original model. Sparc3d proposed a method based on flood fill and optimization to further reduce the loss of precision caused by this conversion, but this introduces self-intersections in the output. Since Sparc3d has not open-sourced any code, we use Dora's method for data processing: we set $\epsilon = 2/\text{resolution}$ for each resolution to extract the surface. This processed data is then used as the GT mesh and input to each method to obtain the reconstruction results. Since the $\epsilon$-level set method results in the loss of object surface details, and this loss is particularly severe at lower resolutions (higher $\epsilon$), our method achieves the best results only at Level 4 on the Watertight remesh results. Furthermore, at Level 1, the low-resolution results appear better than the high-resolution results due to the sheer lack of details that require high resolution to capture.

**Raw mesh.** The comparison on raw meshes best reflects a method's ability to reconstruct geometry. However, as mentioned earlier, methods other than ours and TripoSF's cannot be directly applied to raw meshes, which causes the metrics for these methods to fall significantly behind. To avoid unnecessary misunderstanding, we mark these methods with a asterisk ($*$). Although we do not consider internal viewpoints during training, we do not filter any voxels during inference, allowing our method to still generate objects with internal structures, as shown in Figure 9.

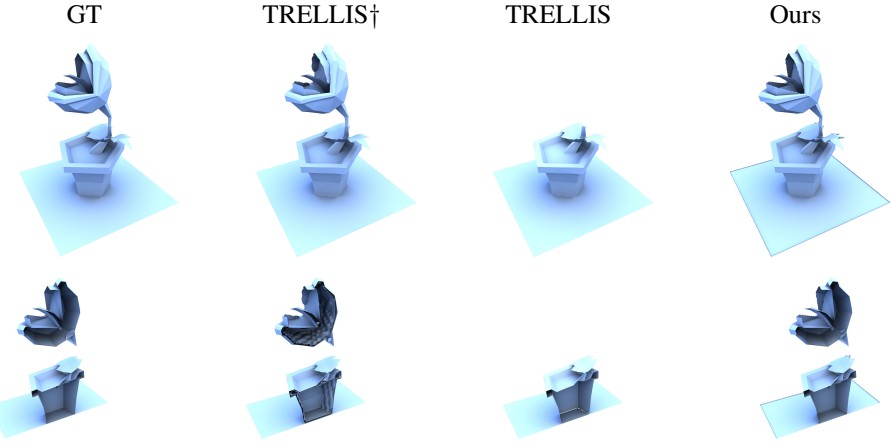

Figure 8: The inner redundant faces generated by TRELLIS.

## B    FAIL CASE

3D VAEs that rely on voxelizied point cloud features inherently struggle to capture thin structures whose thickness is smaller than one voxel. As a result, such structures may be omitted during reconstruction. As illustrated in Figure 10, both our method and TripoSF fail to recover extremely

Table 4: We sample 100K point cloud to measure the Chamfer Distance and F-score on the Dora Benchmark Chen et al. (2025) in different geometry details Levels. Comparison is conducted within each group (Visible Only, Watertight and Raw Mesh respectively).

| | | Chamfer Distance ($10^{-5}$) ↓ | | | | | | | |
| | | L1 | | L2 | | L3 | | L4 | |
| Type | Method | Mean | Std | Mean | Std | Mean | Std | Mean | Std |
|---|---|---|---|---|---|---|---|---|---|
| Visible only | TRELLIS | 1.633 | 1.088 | 2.098 | 2.095 | 5.300 | 68.56 | 4.882 | 11.16 |
| | TRELLIS$^\dagger$ | 1.571 | 0.933 | 1.895 | 1.117 | 2.391 | 1.369 | 3.253 | 4.778 |
| Watertight remeshing | Dora | 1.886 | 0.771 | 2.182 | 0.832 | 2.753 | 1.487 | 3.217 | 1.278 |
| | Direct3d-S2$_{1024}$ | 1.128 | 0.946 | 1.402 | 2.181 | 1.640 | 0.893 | 2.308 | 1.123 |
| | TripoSF$_{512}$ | 1.232 | 0.812 | 1.304 | 0.763 | 1.838 | 1.053 | 2.517 | 1.322 |
| | Ours$_{512}$ | 1.218 | 0.810 | 1.305 | 0.765 | 1.840 | 1.049 | 2.518 | 1.334 |
| | TripoSF$_{1024}$ | 1.243 | 0.825 | 1.321 | 0.741 | 1.844 | 1.007 | 2.426 | 1.184 |
| | Ours$_{1024}$ | 1.242 | 0.824 | 1.324 | 0.741 | 1.785 | 0.898 | 2.258 | 0.906 |
| Raw mesh | TRELLIS* | 266.8 | 2270 | 23.00 | 92.21 | 35.13 | 213.2 | 90.26 | 867.3 |
| | Dora* | 300.5 | 1808 | 225.7 | 1512 | 157.7 | 934.1 | 200.9 | 1447 |
| | Direct3d-S2$_{1024}$* | 335.3 | 1725 | 363.8 | 2161 | 415.2 | 3224 | 425.4 | 2336 |
| | TripoSF$_{512}$ | 1.382 | 0.985 | 1.600 | 1.189 | 2.184 | 1.368 | 3.107 | 2.126 |
| | Ours$_{512}$ | 1.353 | 0.995 | 1.513 | 1.008 | 2.116 | 1.351 | 2.806 | 1.771 |
| | TripoSF$_{1024}$ | 1.315 | 0.937 | 1.456 | 0.936 | 2.007 | 1.197 | 2.431 | 1.390 |
| | Ours$_{1024}$ | 1.294 | 0.886 | 1.429 | 0.873 | 1.901 | 1.011 | 2.264 | 1.055 |

| | | F-score ↑ | | | | | | | |
| | | L1 | | L2 | | L3 | | L4 | |
| Type | Method | Mean | Std | Mean | Std | Mean | Std | Mean | Std |
|---|---|---|---|---|---|---|---|---|---|
| Visible only | TRELLIS | 0.946 | 0.075 | 0.928 | 0.078 | 0.897 | 0.080 | 0.861 | 0.074 |
| | TRELLIS$^\dagger$ | 0.945 | 0.076 | 0.928 | 0.078 | 0.896 | 0.081 | 0.862 | 0.074 |
| Watertight remeshing | Dora | 0.959 | 0.069 | 0.937 | 0.081 | 0.887 | 0.108 | 0.838 | 0.117 |
| | Direct3d-S2$_{1024}$ | 0.974 | 0.048 | 0.969 | 0.055 | 0.941 | 0.074 | 0.887 | 0.106 |
| | TripoSF$_{512}$ | 0.964 | 0.061 | 0.963 | 0.061 | 0.920 | 0.094 | 0.862 | 0.121 |
| | Ours$_{512}$ | 0.965 | 0.061 | 0.963 | 0.061 | 0.920 | 0.093 | 0.862 | 0.122 |
| | TripoSF$_{1024}$ | 0.963 | 0.062 | 0.963 | 0.059 | 0.922 | 0.089 | 0.874 | 0.112 |
| | Ours$_{1024}$ | 0.963 | 0.062 | 0.962 | 0.059 | 0.927 | 0.079 | 0.889 | 0.087 |
| Raw mesh | TRELLIS* | 0.430 | 0.221 | 0.719 | 0.149 | 0.665 | 0.150 | 0.646 | 0.189 |
| | Dora* | 0.379 | 0.077 | 0.408 | 0.079 | 0.337 | 0.068 | 0.292 | 0.084 |
| | Direct3d-S2$_{1024}$* | 0.370 | 0.361 | 0.339 | 0.277 | 0.240 | 0.175 | 0.269 | 0.166 |
| | TripoSF$_{512}$ | 0.951 | 0.078 | 0.939 | 0.089 | 0.892 | 0.115 | 0.831 | 0.146 |
| | Ours$_{512}$ | 0.953 | 0.080 | 0.945 | 0.086 | 0.897 | 0.117 | 0.841 | 0.147 |
| | TripoSF$_{1024}$ | 0.957 | 0.074 | 0.951 | 0.078 | 0.907 | 0.106 | 0.873 | 0.121 |
| | Ours$_{1024}$ | 0.958 | 0.070 | 0.953 | 0.073 | 0.916 | 0.091 | 0.886 | 0.097 |

thin geometric elements. Handling these sub-voxel structures remains an open challenge for voxel-based representations and is an important direction for future work.

Exterior        Cut view

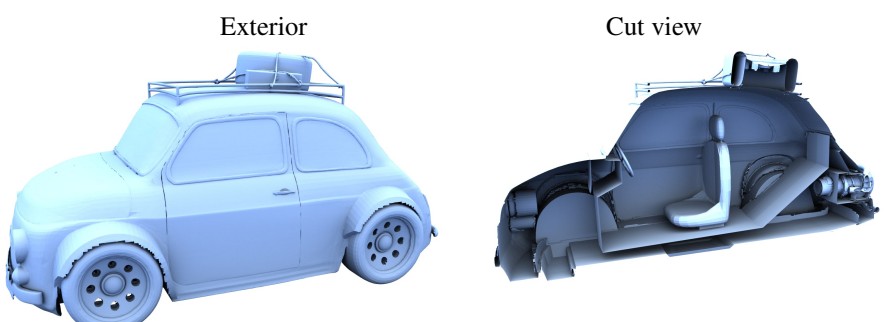

Figure 9: Our method is able to produce shapes with inner structures.

GT          Sparc3D†          Ours

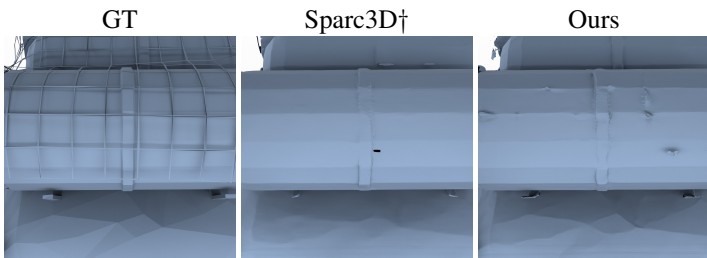

Figure 10: Fail case. Neither our method nor TripoSF can reconstruct extremely thin structures.

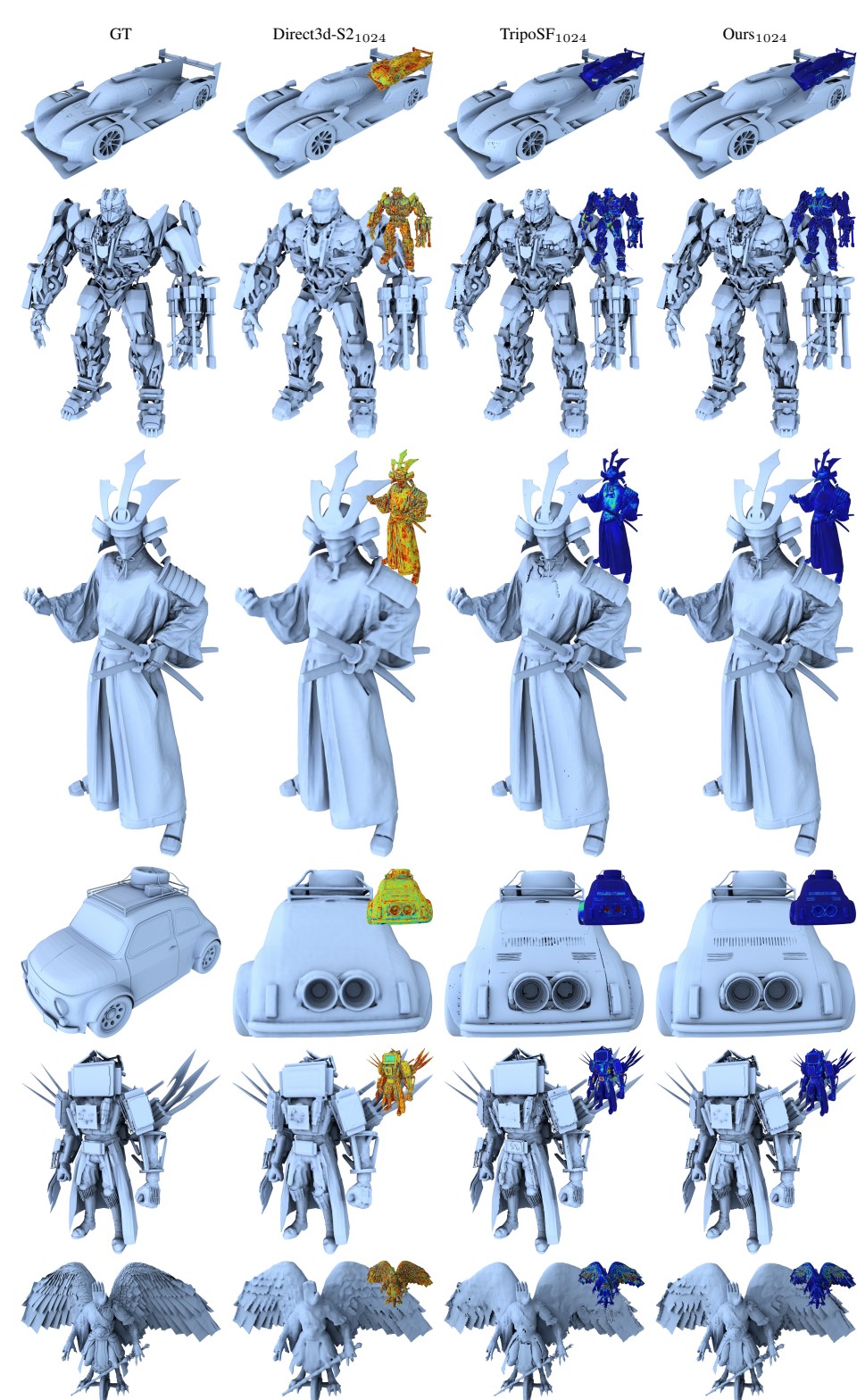

Figure 11: More qualitative comparison of VAE reconstruction between ours, TripoSF and Direct3d-S2 with $1024^3$ resolution. Our approach demonstrate superior performance in reconstructing geometry details.

