# OpenReview forum: "Focusing: View-Consistent Sparse Voxels for Efficient 3D VAE"
_ICLR.cc/2026/Conference — Submitted to ICLR 2026_

### Official Review · Reviewer_CReW · 2025-10-31

**Soundness:** 2
**Presentation:** 2
**Contribution:** 3
**Rating:** 4
**Confidence:** 3

**Summary:**

The paper introduces a 3D VAE (Focusing) designed to improve the efficiency and detail of 3D shape generation. It proposes to mainly address the problem of existing methods to compute all voxels in the camera frustum, which could be irrelevant to the final review, hindering the scalability of high resolution. It introduces two core contributions, including depth-driven voxel carving and adaptive zooming to deal with this problem.

**Strengths:**

1. The contributions of carving and adaptive zooming are logical optimizations. Pruning occluded voxels before the decoding step and zooming to supervise fine-grained details seem to be two effective approaches .
2. The claim of training a $1024^3$ model on < 50GB VRAM is very useful to make high-resolution 3D more accessible.

**Weaknesses:**

The primary weakness is the experimental validation, which lacks sufficient comparison and ablation studies to support the main claims.
1. The authors introduce several components simultaneously, but only compare the final model against the baseline where this work is developed. There is no way to disentangle the effect of each component.
2. Although the paper claims it's more efficient compared to previous works, there is no quantitative evaluation based on it
3. Limited comparison over generative experiments, such as the presentation of Section 4.3 as a validation of VAE's utility, but results are purely qualitative; There are also unsupported claims on regularization.

**Questions:**

1. The comprehensive ablation study could help isolate the individual contributions. For example, it could include the baseline, baseline+depth-driven voxel carving (demonstrating that carving does not affect 3D accuracy), baseline+carving+adaptive zooming (zooming could improve fine-grained details), and the full model.
2. The VRAM comparison or any efficiency quantitative results could help: (a) compare with previous approaches, (b) compare before and after using carving.
3. There is only one comparison of this approach with the TripoSF baseline. Could the model be compared with others, such as the TRELLIS, Sparc3D, mentioned in the paper?
4. For the image-to-3D generation experiments, how does the model quantitatively compare to other generative models (TripoSF, TRELLIS, at least)?
5. For some parameters, such as the depth-carving threshold r = 2 / resolution, what's the model's sensitivity to this hyperparameter? What would happen visually if over-caving or under-caving

---

> ### Author Response · Authors · 2025-11-25
>
> ## Q: The comprehensive ablation study could help identify the individual contributions.
>
>
> A: Thank you for your suggestion. We have added ablation studies in the Table 3 of the revision, showing VRAM usage and training speed under different settings.
>
> ## Q: The VRAM comparison or any efficiency quantitative results
>
> A: Thank you for your suggestion. We have included the relevant table in the revision.
>
>
>
> ## Q: There is only one comparison of this approach with the TripoSF baseline. Could the model be compared with others, such as the TRELLIS, Sparc3D, mentioned in the paper?
>
> A: Unfortunately, Sparc3D has not open-sourced their code or weights, and it has not been formally accepted by any conference or journal. We have added experiments related to Trellis, but we want to highlight that a direct comparison between Trellis and our method is not fair. We provide both the evaluation metrics using Trellis’s original evaluation script and those from our evaluation protocol. Meanwhile, the comparision with Dora-VAE and Direct3d-S2 is also included in the revision.
>
> ## Q: For the image-to-3D generation experiments, how does the model quantitatively compare to other generative models (TripoSF, TRELLIS, at least)?
>
> A: TripoSF has not released their DiT code or weights, so a direct comparison of image-to-3D generation results is not possible. Moreover, recent work focusing on new 3D VAEs does not provide quantitative experiments on image-to-3D generation. In these works, image-to-3D generation is mainly used to verify that the VAE’s latent space satisfies the KL divergence constraint and can produce 3D shapes, rather than functioning as a primary evaluation.
>
>
> ## Q:For some parameters, such as the depth-carving threshold r = 2 / resolution, what's the model's sensitivity to this hyperparameter? What would happen visually if over-caving or under-caving
>
> A: We provide an analysis of different depth-carving thresholds in Figure 4. Under-caving does not affect the output geometry, but it retains more voxels and increases computational cost. Therefore, we recommend using a threshold that is sufficient but not excessive. Over-caving, on the other hand, results in incorrect geometry.

---

### Official Review · Reviewer_RLB5 · 2025-11-02

**Soundness:** 3
**Presentation:** 3
**Contribution:** 3
**Rating:** 8
**Confidence:** 4

**Summary:**

The paper proposes FOCUSING, a 3D VAE that introduces depth-driven voxel carving in the latent space to prune voxels that are inconsistent with a rendered depth map before decoding. This concentrates computation on view-relevant geometry, reduces attention/decoding cost, and enables 1024^3 resolution training with manageable amount of VRAM using a render-based loss plus simple regularizers. To stabilize the number of active voxels, the authors add an adaptive zooming scheme that modifies intrinsics to keep active voxels within a target range. On reconstruction benchmarks, the method outperforms TripoSF at comparable resolutions, and produces strong qualitative image-to-3D results.

**Strengths:**

1. Carving structured latents via depth consistency (as opposed to frustum culling only) is a clear, targeted idea that reduces both decode and render cost. Prior works (TRELLIS SLAT, SparseFlex/TripoSF, etc,) prunes by camera frusta or quotas; carving by per-voxel depth agreement in latent space appears novel in this context and is well-motivated by the render-supervision regime where only visible surfaces receive strong gradients.
2. The proposed efficient latent pruning enables 1024^3 resolution meshes under render-loss supervision, addressing a real bottleneck in scalable 3D VAEs and complements the trend toward structured 3D latents (TRELLIS) and SparseFlex-style meshing. This could potentially be a useful plug-in for render-supervised VAEs and image-to-3D pipelines.
3. The idea is well-motivated. The paper is well-written and clearly presents a concrete algorithm, detailing the loss design and features relevant ablation studies.

**Weaknesses:**

The paper presents a clean and novel idea. I do not have a major concern here, several smaller suggestions and possible improvements:
1. Comparative scope is narrow. Quantitative comparisons focus on TripoSF; there is no direct comparisons with other SLAT VAEs on reconstruction quality. At least Dora-VAE and its followups can be tested for comparison on Dora-Bench. Including more relevant baselines and include CD/F-score across complexity strata (possibly with some case studies) would better demonstrate the gain over previous baselines.
2. Clarity on gradient flow through carving- the voxel-carving mask appears non-differentiable; the paper does not discuss how gradients propagate to pruned voxels, whether there is bias introduced by hard masking, or whether stochasticity/annealing (e.g., soft masks) was explored.
3. Efficiency claims need fuller evidence. The abstract claims <50 GB VRAM for 1024³; however, peak VRAM vs. resolution and batch size are not plotted, nor compared to SparseFlex’s frustum-aware training. A standardized memory/time breakdown (encode/decode/render) against TripoSF/TRELLIS would strengthen the efficiency story.
4. Potential view-bias / consistency issues. Because pruning depends on the current rendered depth, supervision may under-regularize self-occluded or thin structures. A systematic evaluation (multi-view consistency metrics, topology errors, hole rates) and failure case study is absent.

**Questions:**

Please check the weaknesses for details.

---

> ### Author Response · Authors · 2025-11-25
>
> ## Q: Comparative scope is narrow.
>
>
> A: Thank you for your suggestion. In the revision, we have added more comparative experiments to demonstrate the reconstruction performance of our VAE. The CD heatmap visualizations for different methods in Figure 6 show that our method’s can reconstruct fine details better than others.
>
> ## Q: Clarity on gradient flow through carving- the voxel-carving mask appears non-differentiable
>
> A: Discarding part of the output during network training is common; the most recognized example is dropout, which randomly removes sections of the network output to reduce overfitting. Similarly, such pruning operations have been used in Xcubes, TripoSF, and Sparc3D to remove voxels that are far from the surface in the upsampling module. In our encoder, voxels share information through attention mechanisms, which create indirect dependencies between retained and pruned voxels in the latent space. For instance, in the final layer of the encoder, pruned voxels often share similar feature representations with nearby retained voxels. Therefore, removing voxels that are either useless or weakly relevant does not significantly reduce overall performance.
>
> From the results, we see that Trellis uses window-shifted attention—a local attention mechanism—where distant voxels hold less information than nearby ones, effectively serving as a type of soft mask. However, such soft masking does not decrease the number of voxels, which is a key goal of our hard carving strategy.
>
>
> ## Q: Efficiency claims need fuller evidence. The abstract claims <50 GB VRAM for 1024³; however, peak VRAM vs. resolution and batch size are not plotted, nor compared to SparseFlex’s frustum-aware training. A standardized memory/time breakdown (encode/decode/render) against TripoSF/TRELLIS would strengthen the efficiency story.
>
> A: Thank you for your comment. In our experiments, we set the total batch size to 32 and trained using 32 GPUs, resulting in a per-GPU batch size of 1. We have included details on peak VRAM usage in the revision. However, since the code for SparseFlex’s frustum-aware training has not been open-sourced, we cannot report TripoSF’s metrics on our data. We plan to include a detailed memory/time breakdown (encode/decode/render) of our method in a future update.
>
>
> ## Q: Potential view-bias / consistency issues.
>
> Reconstructing thin structures is a known challenge for point-cloud-based 3D VAEs, and both our method and TripoSF have difficulties in such situations. We include examples of these failure cases in the appendix of the revision.
>
> Raw mesh data often contains non-manifold geometries, which makes it difficult to reliably compute the true number of holes. Thus, hole rate and topology errors are hard to measure objectively.
>
> The multi-view consistency metrics mentioned in METeR are mainly intended to measure consistency between pairs of RGB images. However, during inference, our method does not use carving, and the output is a complete 3D model. The reconstruction inherently satisfies multi-view consistency by its design. Therefore, evaluating multi-view consistency metrics on the final results is likely to provide little extra insight.

---

> ### Author Response · Authors · 2025-12-03
>
> Time breakdown in different setting
>
> | | w/o Carving & Zooming | w/o Zooming | Ours ($\alpha_{min}=0.3$) | Ours ($\alpha_{min}=0.6$) | Ours ($\alpha_{min}=0.15$) |
> | :--- | :---: | :---: | :---: | :---: | :---: |
> | Encoding time | 14.8% | 27.0% | 43.7% | 41.1% |  44.6% |
> | Decoding time | 85.2% | 73.0% | 56.3% | 58.9% |  55.4% |

---

### Official Review · Reviewer_j9yh · 2025-11-02

**Soundness:** 3
**Presentation:** 3
**Contribution:** 2
**Rating:** 2
**Confidence:** 4

**Summary:**

This paper focuses on the task of 3D VAE training for 3D object generation. It builds upon and extends the concept of SparseFlex, aiming to further improve training efficiency. The paper introduces two key techniques—depth-based voxel carving and adaptive zooming—which effectively reduce and stabilize the sparse voxels involved in decoding and rendering, thereby decreasing and stabilizing VRAM usage. These techniques also enhance the overall quality of 3D VAE training.

**Strengths:**

1. **Technical Soundness:**
   I find the proposed techniques of voxel carving and adaptive zooming to be both well-motivated and technically sound. Although the ideas represent relatively straightforward engineering improvements, they serve as a valuable complement to SparseFlex.

2. **Clarity and Presentation:**
   The paper is generally well written and easy to follow. It provides sufficient context and is largely self-contained. I also appreciate the well-designed figures, which effectively help readers grasp the core ideas.

3. **Experimental Results:**
   The paper successfully demonstrates its superiority in terms of fitting error.

**Weaknesses:**

While the paper is relatively straightforward and engineering-oriented, I am not against that—I believe the idea is a meaningful and well-motivated complement to existing work. However, my major concern lies in the completeness of the evaluation. As a scientific paper submitted to ICLR, the current version’s experimental validation is far from sufficient. I understand that the authors may have completed the paper in a hurry and plan to add more experiments during the *long-period* rebuttal stage of ICLR. Nevertheless, I personally feel that this practice is unfair to other authors who submitted fully developed experiments before the deadline. As a result, I assign a **reject** rating here primarily due to the paper’s **incompleteness** and **fairness concerns**.

1. **Lack of Efficiency Analysis:**
   While the main claim is that the proposed techniques improve efficiency, there is no direct analysis of training or inference time, VRAM usage, or the resolution of supervision images. The only provided experiment concerns the VAE’s fitting quality. As a scientific paper, we not only care about fitting error but also about how efficiency-related metrics are affected by the proposed techniques. Without these analysis, the claim of efficiency improvement remains unsubstantiated.

   For instance, why does the proposed VAE achieve better fitting error? Is it because the reduced VRAM usage allows training with higher-resolution supervision images? Such quantitative evidence and analysis are essential. Furthermore, if the proposed method’s main advantage is efficiency, one might question whether similar improvements could be achieved simply by using smaller batch sizes and more GPUs.

2. **Missing Ablation Studies:**
   The paper introduces two main techniques—depth-based voxel carving and adaptive zooming—as well as additional loss terms. However, no ablation studies are provided. For a scientific paper, extensive ablations are necessary to understand the contribution of each component and loss function to VRAM usage, training/inference time, and fitting quality. Without such experiments, it is difficult to attribute the observed performance improvements to specific innovations.

**Questions:**

0. **Limited Decoding Capability Compared to Trellis:**
   The original *Trellis* framework supports multiple decoding heads, such as decoding 3D Gaussians for texture generation. In contrast, the proposed method appears to decode only 3D meshes. How does the method handle mesh textures, and is there a plan to extend the approach to support texture decoding as in Trellis?

1. **Unexplained Term in Equation (2):**
   The loss term ( L_{\text{occ}} ) in Equation (2) is not explained. Please provide a clear definition and motivation for this term.

2. **Lack of Clarity in Lines 317–320:**
   The description of the self-pruning upsampling module is not self-contained and requires more detailed explanation. Additionally, the meaning of ( O_{\text{pred}} ) should be clarified.

3. **Missing Symbol Definitions in Equation (6):**
   Several symbols in Equation (6) are not defined. Please include explanations for all variables used.

4. **Ambiguous Description in Line 348:**
   The phrase “inject them into a Diffusion Transformer (DiT) via cross-attention” is unclear. From my understanding, the method trains an image-conditioned rectified flow model, rather than injecting image features into the latent space prior to rectified flow training. Please clarify this process.

5. **Clarification Needed in Line 369:**
   Why is the number 518 chosen? Please justify this specific design choice.

6. **Unsubstantiated Claim in Line 378:**
   The statement “Our method demonstrates results comparable to TripoSF’s 1024-resolution output even at 512 resolution and captures geometry details better at 1024 resolution” is not clearly supported by quantitative or qualitative evidence. Please provide corresponding results or analysis to substantiate this claim.

---

> ### Author Response · Authors · 2025-11-25
>
> ## Q: The idea is a meaningful and well-motivated complement to existing work. However, the evaluation is incompletment.
>
> A: We sincerely thank the reviewer for the positive assessment that our idea is  "meaningful", "well-motivated", and a valuable complement to existing work. We also appreciate the reviewer's concern regarding the completeness of the evaluation.
>
> We would like to clarify several points regarding the evaluation setup and the role of the discussion period:
>
> 1) Why the initial submission only compared against TripoSF. As explained above, most recent approaches are either not publicly available (e.g., Sparc3D), released only at significantly lower resolutions (e.g., Trellis and Dora-VAE), or require watertight meshes as input (e.g., Direct3D-S2). Under such mismatched conditions, a head-to-head comparison would not be fair or informative. TripoSF is the only method that allows a fully aligned experimental setting. We also add a new table (Table 1) to better illustrate the differences among the various methods.
>
> 2) **The original submission was complete with respect to the method's core claims**. The initial version already included complete VAE reconstruction experiments and image-to-3D generation results to confirm the model's main functions and its latent-space design.
>
> 3) The discussion period aims to strengthen the paper. The ICLR author and reviewer guidelines clearly encourage authors to address reviewer feedback during the rebuttal and discussion phase. This includes adding experiments or clarifying technical points. The additional results we provide were conducted in direct response to reviewer questions. They do not change the core method; instead, they provide the extra empirical evidence asked for by the reviewers.
>
> Following the reviewers' suggestions, we have added: (1) comparisons with Trellis, Dora-VAE, and Direct3D-S2, (2) ablation studies isolating carving, zooming, and warm-up behavior, and (3) efficiency analysis including VRAM usage and runtime under different settings.
>
> We hope that these clarifications, along with the improved experimental section in the updated paper, will help the reviewer reassess the submission based on its technical contributions and the additional empirical evidence presented during the discussion phase.
>
> ## Q: Lack of Efficiency Analysis
>
> A: Thank you for your feedback.
>
> Regarding training efficiency, we state in Section 4.1 that our method trains at 512 resolution in just 2 days using 32 A800 GPUs, and at 1024 resolution in 2 days using 32 A100 GPUs. In contrast, TripoSF reports using 64 A100 GPUs but does not disclose its training time and does not release official training or preprocessing code. Trellis uses a similar pipeline and takes about one week to train.
>
> Regarding supervision image resolution, we also mention in Section 4.1 that our supervision images have a resolution of 518×518, which matches that of Trellis. Because we adjust the camera’s field of view during training, we do not need to crop or resize images, which keeps the supervision resolution consistent without extra effort.
>
> In terms of inference, our framework has a design similar to TripoSF, so the speed of inference is comparable and is not the main focus of our efficiency claims.
>
> To further support our claims of efficiency, we have included an ablation study in the revised manuscript (see Table 3) that reports VRAM consumption and training speed under different configurations (e.g., different zoom rates), addressing the requested efficiency metrics.
>
>
>
> ## Q: Missing Ablation Studies
>
> A: Thank you for your suggestion. In the revision, we have added a table showing the effect of depth-based voxel carving and adaptive zooming on VRAM usage and running time. We have also included a qualitative experiment in Figure 11, demonstrating how the TV loss affects the results, particularly how it helps in reducing holes.

---

> ### Author Response · Authors · 2025-11-25
>
> ## Q: Limited Decoding Capability Compared to Trellis
>
> A: Although Trellis uses a 3D Gaussian Splatting (3DGS) decoder to generate textures, the realism is still limited (e.g., it does not support PBR materials). Moreover, Trellis does not output geometry and texture together; instead, it uses a separate decoder to create the geometry and then bakes textures from the 3DGS renderings. Given the distortions in 3DGS renderings, this approach has some limitations. Currently, both mainstream open-source and closed-source 3D AIGC pipelines primarily generate geometry and texture separately. Our work focuses on geometry modeling. Textures can be created later using multi-view 2D image generation methods and then mapped back onto the mesh.
>
>
> ## Q: The description of the self-pruning upsampling module is not self-contained and requires more detailed explanation.
>
> A:
> The self-pruning upsampling module was introduced by Xcubes to enhance the resolution of VAE outputs. To prevent the voxel count from increasing too much during upsampling, Xcubes uses an additional MLP to predict the probability $O_{pred}$ for each upsampled voxel and then removes voxels that are far from the surface. The loss term $L_{occ}$ is calculated by comparing the predicted occupancy with the upsampled ground-truth occupancy. The module provides the probability for each voxel to be near the surface. This module has been adopted in TripoSF and SparC3D to produce high-resolution results. However, inaccurate occupancy predictions can create gaps in the output, which negatively impact the convergence of the rendering loss during training. Therefore, similar to Trellis, we do not use the self-pruning module in our network and retain only the upsampling component.
>
>
> ## Q: Missing Symbol Definitions in Equation (6)
>
> A: Thank you for your suggestion. We will include the relevant descriptions in the revision. $\Delta_x^2(V, d)$ denotes the squared difference between the value of the $d$th channel in voxel $v := (i; j; k)$ and the $d$th value in voxel $(i + 1; j; k)$. This can be similarly extended to $\Delta_y^2(V, d)$ and $\Delta_z^2(V, d)$. We apply the TV term above to the SDF grid, denoted by $\mathcal{L}_{tv}$, which encourages a continuous and compact geometry.
>
> ## Q: Ambiguous Description in Line 348
>
> A: Thanks for pointing out this issue. We will clarify it in the revision. Specifically, we use image features extracted by DINOv2 as a condition to train a rectified flow model for generation.
>
> ## Q: Clarification Needed in Line 369: Why is the number 518 chosen? Please justify this specific design choice.
>
>
> A: 518 is the input resolution of DINOv2, we did not make any special resolution settings.
>
>
>
> ## Q: The statement “Our method demonstrates results comparable to TripoSF’s 1024-resolution output even at 512 resolution and captures geometry details better at 1024 resolution” is not clearly supported by quantitative or qualitative evidence.
>
> A: Thank you for your suggestion. We have demonstrated in Figure 6 that our results at 512 resolution are better than TripoSF’s results at 1024 resolution in some cases (e.g., armor details of the model). The quantitative results are not clearly evident because the Dora benchmark contains only a limited number of high-fidelity models in Level 4. We are collecting additional 3D models to better showcase the advantages of our method.
>
> Accordingly, we have revised the original statement to: "Our method demonstrates results comparable to TripoSF’s 1024 resolution output. In some high-fidelity models, our 512 resolution result captures geometry details better than TripoSF’s 1024 resolution result, as shown in Figure 6." This revised wording is more precise and better reflects the evidence.

---

### Official Review · Reviewer_YxmK · 2025-11-04

**Soundness:** 2
**Presentation:** 3
**Contribution:** 2
**Rating:** 4
**Confidence:** 5

**Summary:**

FOCUSING targets high-fidelity yet efficient 3D generation by training a 3D VAE with render-based supervision to reconstruct meshes from sparse voxel latents, enabling 1024³ results with modest memory. It is adapted based on TripoSF. Its core idea is depth-driven voxel carving in the latent space that prunes voxels inconsistent with the rendered depth before decoding, concentrating learning on view-relevant geometry. An adaptive “zooming” strategy further adjusts camera intrinsics to keep active-voxel counts within a target range, stabilizing VRAM and sharpening details. On Objaverse-XL–scale training, the method cuts VRAM to ≲50 GB for 1024³ and improves Chamfer Distance and F-score over TripoSF at matched resolutions, with visual detail comparable to higher-res baselines. Overall, the work shows that local, view-consistent sparsity contributes to higher-resolution and more efficient 3D VAEs.

**Strengths:**

- Better efficiency and higher resolution than TripoSF for a 3D-VAE.
- Practical training design: render-based supervision (depth/normal/mask + perceptual losses), sparse-voxel TV, and a short TSDF warm-up stabilize learning and sharpen details.
- Adaptive “zooming” to keep active-voxel counts in a target range, improving memory predictability and fine-structure fidelity.
- The paper is clearly written, and the figures and formulas are clean.

**Weaknesses:**

- Internal structure is largely neglected: the efficiency comes from more selective surface sampling, but an informative 3D-VAE ideally encodes both exterior and interior—methods like SPARC3D highlight this. The trade-off may be suboptimal for full 3D reconstruction use cases.
- Limited novelty: much of the pipeline builds on TripoSF; the main additions (carving + zooming) feel like incremental efficiency improvements.
- Insufficient baselines: Only VAE reconstruction comparison (Table 1) to TripoSF. Missing head-to-head results against other single-image 3D reconstruction methods (e.g., Sparc3D). Even within VAE reconstruction, more baselines and Dora metrics should be included.
- Marginal quantitative gains: the reported improvements over TripoSF appear small ( <0.1σ in the table), which weakens the practical significance of the contribution.
- No qualitative ablations isolating design choices (carving thresholding, zoom schedule, TSDF warm-up length) to show when each component helps or hurts.

**Questions:**

- Please include a clear VRAM and wall-clock table (peak and average) across resolutions and batch sizes, alongside TripoSF and any additional baselines.
- How sensitive are results to the carving threshold/heuristics and zoom schedule? Provide qualitative ablations on thin parts and hollow objects.

Minor issues:

- L42: there is a missing space before “We.”
- Duplicate references of PointNet

---

> ### Author Response · Authors · 2025-11-25
>
> ## Q: Internal structure is largely neglected
>
> A: This is a misunderstanding. Our method can generate internal geometry because we employ a TripoSF-style backbone that preserves all voxels during inference. Although we do not use internal viewpoints during training -- because modifying the near plane can introduce instability -- internal voxels remain intact in the latent representation. As long as these voxels are not removed during carving, our method reconstructs internal geometry correctly.
>
> All comparisons are conducted on the Dora benchmark without removing internal structures, and our method achieves higher accuracy than TripoSF on raw meshes that contain internal structures. We also include visual examples in the Figure 9 in revised paper demonstrating successful reconstruction of internal geometry.
>
>
> ## Q: Limited novelty
>
> A: We appreciate that TripoSF introduced a sparse version of FlexibleCubes, which significantly reduces GPU memory usage during mesh extraction. However, frustum-based cropping often results in rendered outputs that contain many separate components rather than a single unified surface. Our experiments further show that most invisible regions receive almost no gradient updates during training yet still occupy substantial memory, and frustum-based carving cannot reliably remove these voxels.
>
> For this reason, we adopt a visibility-based carving strategy that removes voxels from locations that are truly invisible. Our experiments show that this pruning can be safely applied in the latent space, leading to substantial memory savings. In TripoSF, the visibility ratio $\alpha$ controls the proportion of active voxels in SparseFlex. At 512 resolution, TripoSF uses $\alpha=0.3$ for most of their experiments, requring 45.6 GB of GPU memory; even with $\alpha=0.1$, the memory requirement remains above 40.1 GB of memory.
>
> In contrast, our method uses the visibility ratio $\alpha$ to control the zooming window, defined as  the ratio of voxels inside the zoomed field of view to those in the original view. We randomly sample $\alpha\in[\alpha_{min}, \alpha_{max}$] to balance both near and far viewpoints during training. With $\alpha_{min} = 0.1$ and $\alpha_{max} = 0.3$, our method requires only 32 GB of GPU memory. Reducing $\alpha_{max}$ to 0.15 further lowers memory use to 28 GB. Moreover, unlike strategies that rely on adjusting near and far clipping planes, our visibility-guided zooming maintains surface continuity and avoids fragmented renderings.
>
> **In summary, the simplicity of the proposed strategies is a key strength:**
> i) They are separate from the backbone architecture, easy to integrate, and require no new modules or heavy machinery. ii) They yield direct and substantial benefits, removing redundant voxels, reducing memory footprint, and enabling higher-resolution training. iii) They are highly reproducible, lowering the barrier for future research to adopt and extend the ideas. Given these strengths, we believe that our method has great potential to be a practical and widely used part of future 3D generative pipelines. This approach can enable more efficient training, higher-resolution modeling, and easier integration into both existing and new 3D VAE frameworks.

---

> ### Author Response · Authors · 2025-11-25
>
> ## Q: Insufficient baselines
>
> A: We would like to clarify that our initial submission did not include comparisons with certain methods because their implementations are either not available or only offered at significantly lower resolutions than ours. Using such baselines would lead to mismatched experimental conditions and possibly misleading comparisons.
>
> For example, although we also intended to compare with Sparc3D, there is currently no publicly available implementation, and their paper lacks important implementation details. These missing components make it difficult to reproduce the pipeline and train the network.
>
> In the revision, we have included comparisons with Trellis, Dora-VAE and Direct3D. However, only the 256-resolution version of Dora-VAE has been publicly released, which leads to significantly lower performance compared to our method and TripoSF. In addition, Trellis removes interior structures through post-processing, while Dora discards internal geometry during data preprocessing; both operations further decrease their quantitative metrics. To reduce any potential misunderstandings from differences in preprocessing, we also report evaluation metrics using meshes processed with Dora's pipeline as the ground truth in the updated paper. Trellis is also evaluated using the visibility-based Chamfer Distance protocol described in its original paper. Meanwhile, since Direct3D also requires a watertight mesh as input, we evaluate it in a similar way to Dora-VAE.
>
> That said, TripoSF is the only method that enables a **fair, apples-to-apples comparison** with ours. Comparisons with Trellis or Dora-VAE are biased because of their different preprocessing methods and resolution limits, and they may be viewed as unfair experimental setups.
>
>
> ## Q: Marginal quantitative gains
>
> A: The models in the Dora benchmark are not carefully chosen high-quality assets, making it difficult to fully demonstrate the benefits of our method in detailed reconstruction. To demonstrate the practical importance of our improvements, we plan to include an extra benchmark made up of high-quality, precise models in future experiments. Moreover, the qualitative comparisons in Figure 6, which feature high-detail models, clearly show our method’s greater ability to capture complex geometric details.
>
> ## Q: No qualitative ablations
>
> A:
> 1) Carving thresholding. We use a voxelized representation of the mesh as input to our network. During depth carving, we trim voxels based on the centers of the voxels. However, some voxels intersecting the rendered surface may have their centers located behind the surface due to discretization. To address this, we introduce a carving threshold. We set this threshold to 2/resolution , which corresponds to the width of one voxel. This is sufficient to include all voxels related to the visible surface layer. In Figure 4 of our revision, we present a comparison of different threshold values. When the threshold is set to 0, useful voxels near surface boundaries may be mistakenly discarded, leading to visual artifacts. Reducing the threshold further to −2/resolution leads to significant surface damage. Conversely, increasing the threshold to 4/resolution yields no significant geometric improvement but slightly increases the number of retained voxels.
>
> 2) Zoom schedule. During training, we randomly sample a zoom factor uniformly from the range [$\alpha_{min}$,$\alpha_{max}$] for each iteration. In our experiments, we set $\alpha_{min}=0.1$ and $\alpha_{max}=0.3$, which matches the setting of TripoSF. Table 3 shows the memory use and runtime for different $\alpha_{max}$ values. A higher $\alpha_{max}$ may increase GPU memory usage, while a lower $\alpha_{max}$ requires more training steps to reach convergence.
>
> 3) TSDF warm-up length. We initialize the weights of the network’s output layer to zero. In methods like Trellis, an initial mesh is created by subtracting a bias from the network output and combining it with a background SDF value of 1. However, in our sparse voxel architecture, we cannot directly pad a full background with SDF = 1. To address this, we employ a TSDF warm-up phase to gradually set the network weights. We set the warm-up length to 12,000 steps, which only affects the very early stage of training and has a minor impact on the final results. Using TSDF supervision throughout the entire training process can create problems due to incorrect surface normals in intermediate reconstructions.

---

### Author Response · Authors · 2025-11-25

We thank the reviewers for their detailed and constructive feedback. Below, we address key concerns and clarify several misunderstandings.

# Reviewer YxmK: No internal structures
This is a misunderstanding. Our method can generate internal geometry because we employ a TripoSF-style backbone that preserves all voxels during inference. Although we do not use internal viewpoints during training -- because modifying the near plane can introduce instability -- internal voxels remain intact in the latent representation. As long as these voxels are not removed during carving, our method can successfully reconstruct internal geometry. Our comparisons are conducted on the Dora benchmark without removing internal structures, and the results show that our method achieves higher accuracy than TripoSF on raw mesh data that contains internal structures. We also provide visual examples in the updated paper demonstrating our ability to reconstruct internal geometry.

# Reviewer YxmK, j9yh, RLB5, CReW: Lack of comparisons with recent methods
We would like to clarify that our initial submission did not include comparisons with several recent methods because their implementations are either unavailable (e.g., Sparc3D, whose paper omits essential details required to reproduce its preprocessing pipeline) or only released at significantly lower resolutions than ours (e.g., Trellis and Dora-VAE, both limited to 256 resolution). In addition, Direct3D-S2 supports 1024 resolution but requires watertight meshes as input, which is incompatible with our evaluation setting. Under such mismatched conditions, comparisons would be unfair or misleading.

In the revised version, we have included comparisons with Trellis and Dora-VAE using their publicly released 256-resolution models, as well as with Direct3D using its publicly available 1024-resolution configuration. For Direct3D, we preprocess the meshes using Dora’s watertight repair pipeline to meet its input requirements.

We also include a comparison table that summarizes key differences among these methods for clarity.


# Reviewer YxmK, j9yh, RLB5, CReW : Lack of ablation studies
Thank you for the suggestion. In the revised version, we have added ablation studies on both the zooming and carving components, reporting their effects on memory consumption and runtime.

# Reviewer YxmK: Comparison with TripoSF
We appreciate that TripoSF introduced a sparse version of FlexibleCubes, which significantly reduces GPU memory usage during mesh extraction. However, frustum-based cropping often results in rendered outputs containing many disconnected components rather than a single coherent surface. Our experiments further show that most invisible regions receive almost no gradient updates during training yet still occupy substantial memory, and frustum-based carving cannot reliably remove these voxels.

For this reason, we adopt a visibility-based carving strategy that removes voxels at truly invisible locations. Our experiments show that this pruning can be safely applied in the latent space, leading to substantial memory savings. In TripoSF, the visibility ratio $\alpha$ controls the proportion of active voxels in SparseFlex. At 512 resolution, TripoSF uses $\alpha=0.3$ for most of their experiments, requring 45.6 GB of GPU memory; even with $\alpha=0.1$, the memory requirement remains above 40.1 GB of memory.

In contrast, our method uses the visibility ratio $\alpha$ to control the zooming window, defined as  the ratio of voxels inside the zoomed field of view to those in the original view. We randomly sample $\alpha\in[\alpha_{min}, \alpha_{max}$ to balance both near and far viewpoints during training. With $\alpha_{min} = 0.1$ and $\alpha_{max} = 0.3$, our method requires only 32 GB of GPU memory. Reducing $\alpha_{max}]$ to 0.15 further decreases memory usage to 28 GB. Moreover, unlike strategies that rely on adjusting near and far clipping planes, our visibility-guided zooming maintains surface continuity and avoids fragmented renderings.

# Simplicity represents a key strength
The simplicity of the proposed strategies is not a limitation but a core advantage: i) They are orthogonal to the backbone architecture, easy to integrate, and require no new modules or heavy machinery. ii) They yield direct and substantial benefits, removing redundant voxels, reducing memory footprint, and enabling higher-resolution training. iii) They are highly reproducible, lowering the barrier for future research to adopt and extend the ideas. Given these strengths, we believe that our method has a great potential to serve as a practical and widely applicable component in future 3D generative pipelines, enabling more efficient training, higher-resolution modeling, and easier integration into both existing and emerging 3D VAE frameworks.

In the following, we provide point-to-point responses to the reviewers' questions.

---

### Author Response · Authors · 2025-11-25
**F-score in Dora benchmark**

| Type | Method | L1 Mean | L1 Std | L2 Mean | L2 Std | L3 Mean | L3 Std | L4 Mean | L4 Std |
| :---: | :--- | :---: | :---: | :---: | :---: | :---: | :---: | :---: | :---: |
| Visible only | Trellis | 0.946 | 0.075 | 0.928 | 0.078 | 0.897 | 0.080 | 0.861 | 0.074 |
| | Trellis$^\dagger$ | 0.945 | 0.076 | 0.928 | 0.078 | 0.896 | 0.081 | 0.862 | 0.074 |
| Watertight remeshing | Dora | 0.959 | 0.069 | 0.937 | 0.081 | 0.887 | 0.108 | 0.838 | 0.117 |
| | Direct3d-S2$_{1024}$ | $\mathbf{\color{red}0.974}$ | $\mathbf{\color{red}0.048}$ | $\mathbf{\color{red}0.969}$ | $\mathbf{\color{red}0.055}$ | $\mathbf{\color{red}0.941}$ | $\mathbf{\color{red}0.074}$ | $\mathbf{\color{orange}0.887}$ | $\mathbf{\color{orange}0.106}$ |
| | TripoSF$_{512}$ | $\mathbf{\color{blue}0.964}$ | $\mathbf{\color{orange}0.061}$ | $\mathbf{\color{orange}0.963}$ | $\mathbf{\color{blue}0.061}$ | 0.920 | 0.094 | 0.862 | 0.121 |
| | Ours$_{512}$ | $\mathbf{\color{orange}0.965}$ | $\mathbf{\color{orange}0.061}$ | $\mathbf{\color{orange}0.963}$ | $\mathbf{\color{blue}0.061}$ | 0.920 | 0.093 | 0.862 | 0.122 |
| | TripoSF$_{1024}$ | 0.963 | $\mathbf{\color{blue}0.062}$ | $\mathbf{\color{orange}0.963}$ | $\mathbf{\color{orange}0.059}$ | $\mathbf{\color{blue}0.922}$ | $\mathbf{\color{blue}0.089}$ | $\mathbf{\color{orange}0.874}$ | $\mathbf{\color{blue}0.112}$ |
| | Ours$_{1024}$ | 0.963 | $\mathbf{\color{blue}0.062}$ | $\mathbf{\color{blue}0.962}$ | $\mathbf{\color{orange}0.059}$ | $\mathbf{\color{orange}0.927}$ | $\mathbf{\color{orange}0.079}$ | $\mathbf{\color{red}0.889}$ | $\mathbf{\color{red}0.087}$ |
| Raw mesh | Trellis* | 0.430 | 0.221 | 0.719 | 0.149 | 0.665 | 0.150 | 0.646 | 0.189 |
| | Dora* | 0.379 | 0.077 | 0.408 | 0.079 | 0.337 | 0.068 | 0.292 | 0.084 |
| | Direct3d-S2$_{1024}$* | 0.370 | 0.361 | 0.339 | 0.277 | 0.240 | 0.175 | 0.269 | 0.166 |
| | TripoSF$_{512}$ | 0.951 | $\mathbf{\color{blue}0.078}$ | 0.939 | 0.089 | 0.892 | $\mathbf{\color{blue}0.115}$ | 0.831 | $\mathbf{\color{blue}0.146}$ |
| | Ours$_{512}$ | $\mathbf{\color{blue}0.953}$ | 0.080 | $\mathbf{\color{blue}0.945}$ | $\mathbf{\color{blue}0.086}$ | $\mathbf{\color{blue}0.897}$ | 0.117 | $\mathbf{\color{blue}0.841}$ | 0.147 |
| | TripoSF$_{1024}$ | $\mathbf{\color{orange}0.957}$ | $\mathbf{\color{orange}0.074}$ | $\mathbf{\color{orange}0.951}$ | $\mathbf{\color{orange}0.078}$ | $\mathbf{\color{orange}0.907}$ | $\mathbf{\color{orange}0.106}$ | $\mathbf{\color{orange}0.873}$ | $\mathbf{\color{orange}0.121}$ |
| | Ours$_{1024}$ | $\mathbf{\color{red}0.958}$ | $\mathbf{\color{red}0.070}$ | $\mathbf{\color{red}0.953}$ | $\mathbf{\color{red}0.073}$ | $\mathbf{\color{red}0.916}$ | $\mathbf{\color{red}0.091}$ | $\mathbf{\color{red}0.886}$ | $\mathbf{\color{red}0.097}$ |

---

### Author Response · Authors · 2025-11-25
**Chamfer Distance in Dora benchmark**

We present additional comparison results of our method with more approaches on the Dora benchmark as noted by the reviewers. But as we argue in the paper, other methods are not directly applicable for comparison on raw meshes (i.e., the original GT meshes), we mark all these methods with an asterisk (*) to avoid confusion. Additionally, we provide experimental results using Dora’s watertight remeshed meshes as the GT meshes. For different resolutions, we use different eps values (specifically, eps=2/resolution   as suggested by the authors of Dora) to perform watertightification and obtain the GT watertight meshes.

1) Dora-VAE only provides pretrained weights at 256 resolution, so we evaluate Dora-VAE at 256 resolution.

2) Trellis cannot take the remeshed watertight meshes as input and is therefore excluded from direct comparison. However, we evaluate Trellis using the visibility-aware point sampling strategy described in the Trellis paper. Results labeled with "$^\dagger$ " denote evaluations without Trellis’s post-processing step. Although Trellis’s post-processing can remove internal, invisible face fragments, it may also erroneously eliminate valid geometry, leading to degraded quantitative metrics. It is worth noting that Trellis’s evaluation protocol does not consider internal invisible fragments, but in practical applications, such post-processing is always required.

3) Direct3d-S2 is evaluated using the official 1024-resolution weights. However, since Direct3d-S2 requires watertight meshes as input, it only produces valid results on the watertight remeshed data. Dora’s remeshing algorithm produces an eps-expanded version of the target surface, which loses fine details from the original model. Direct3d-S2’s overall metrics appear higher than ours. Nevertheless, on Level 4 (the most complex) data, our method still achieves the best quantitative results.


| Type | Method | L1 Mean | L1 Std | L2 Mean | L2 Std | L3 Mean | L3 Std | L4 Mean | L4 Std |
| :---: | :--- | :---: | :---: | :---: | :---: | :---: | :---: | :---: | :---: |
| Visible only | Trellis | 1.633 | 1.088 | 2.098 | 2.095 | 5.300 | 68.56 | 4.882 | 11.16 |
| | Trellis$^\dagger$ | 1.571 | 0.933 | 1.895 | 1.117 | 2.391 | 1.369 | 3.253 | 4.778 |
| Watertight remeshing | Dora | 1.886 | $\mathbf{\color{red}0.771}$ | 2.182 | 0.832 | 2.753 | 1.487 | 3.217 | 1.278 |
| | Direct3d-S2$_{1024}$ | $\mathbf{\color{red}1.128}$ | 0.946 | 1.402 | 2.181 | $\mathbf{\color{red}1.640}$ | $\mathbf{\color{red}0.893}$ | $\mathbf{\color{orange}2.308}$ | $\mathbf{\color{orange}1.123}$ |
| | TripoSF$_{512}$ | $\mathbf{\color{blue}1.232}$ | $\mathbf{\color{blue}0.812}$ | $\mathbf{\color{red}1.304}$ | $\mathbf{\color{orange}0.763}$ | $\mathbf{\color{blue}1.838}$ | 1.053 | 2.517 | 1.322 |
| | Ours$_{512}$ | $\mathbf{\color{orange}1.218}$ | $\mathbf{\color{orange}0.810}$ | $\mathbf{\color{orange}1.305}$ | $\mathbf{\color{blue}0.765}$ | 1.840 | 1.049 | 2.518 | 1.334 |
| | TripoSF$_{1024}$ | 1.243 | 0.825 | $\mathbf{\color{blue}1.321}$ | $\mathbf{\color{red}0.741}$ | 1.844 | $\mathbf{\color{blue}1.007}$ | $\mathbf{\color{blue}2.426}$ | $\mathbf{\color{blue}1.184}$ |
| | Ours$_{1024}$ | 1.242 | 0.824 | 1.324 | $\mathbf{\color{red}0.741}$ | $\mathbf{\color{orange}1.785}$ | $\mathbf{\color{orange}0.898}$ | $\mathbf{\color{red}2.258}$ | $\mathbf{\color{red}0.906}$ |
| Raw mesh | Trellis* | 266.8 | 2270 | 23.00 | 92.21 | 35.13 | 213.2 | 90.26 | 867.3 |
| | Dora* | 300.5 | 1808 | 225.7 | 1512 | 157.7 | 934.1 | 200.9 | 1447 |
| | Direct3d-S2$_{1024}$* | 335.3 | 1725 | 363.8 | 2161 | 415.2 | 3224 | 425.4 | 2336 |
| | TripoSF$_{512}$ | 1.382 | $\mathbf{\color{blue}0.985}$ | 1.600 | 1.189 | 2.184 | 1.368 | 3.107 | 2.126 |
| | Ours$_{512}$ | $\mathbf{\color{blue}1.353}$ | 0.995 | $\mathbf{\color{blue}1.513}$ | $\mathbf{\color{blue}1.008}$ | $\mathbf{\color{blue}2.116}$ | $\mathbf{\color{blue}1.351}$ | $\mathbf{\color{blue}2.806}$ | $\mathbf{\color{blue}1.771}$ |
| | TripoSF$_{1024}$ | $\mathbf{\color{orange}1.315}$ | $\mathbf{\color{orange}0.937}$ | $\mathbf{\color{orange}1.456}$ | $\mathbf{\color{orange}0.936}$ | $\mathbf{\color{orange}2.007}$ | $\mathbf{\color{orange}1.197}$ | $\mathbf{\color{orange}2.431}$ | $\mathbf{\color{orange}1.390}$ |
| | Ours$_{1024}$ | $\mathbf{\color{red}1.294}$ | $\mathbf{\color{red}0.886}$ | $\mathbf{\color{red}1.429}$ | $\mathbf{\color{red}0.873}$ | $\mathbf{\color{red}1.901}$ | $\mathbf{\color{red}1.011}$ | $\mathbf{\color{red}2.264}$ | $\mathbf{\color{red}1.055}$ |

---

### Author Response · Authors · 2025-11-25
**Training speed and GPU memory cost comparisons**

We present the training speed and VRAM usage under different experimental configurations at 512 resolution in A100. Our method uses the visibility ratio $\alpha$ to control the zooming window, defined as  the ratio of voxels inside the zoomed field of view to those in the original view. We randomly sample $\alpha\in[\alpha_{min}, \alpha_{max}]$ to balance both near and far viewpoints during training.

| | w/o Carving & Zooming | w/o Zooming | Ours ($\alpha_{max}$=0.3) | Ours ($\alpha_{max}$=0.6) | Ours ($\alpha_{max}$=0.15) |
| :---: | :---: | :---: | :---: | :---: | :---: |
| Training Speed (steps/h) | 2652 | 5337 | 8640 | 7590 | 9238 |
| GPU Memory Cost (GB) | 64 | 50 | 32 | 35 | 28 |

---

### Author Response · Authors · 2025-12-03
**Experiments update**

We have compiled the results of the ablation study at $1024$ resolution:

| | w/o Carving & Zooming | w/o Zooming | Ours ($\alpha_{min}=0.3$) | Ours ($\alpha_{min}=0.6$) | Ours ($\alpha_{min}=0.15$) |
| :--- | :---: | :---: | :---: | :---: | :---: |
| Training Speed (steps/h) | OOM | 510 | 892 | 657 | 1003 |
| GPU Memory Peak (GB) | OOM | 78 | 56 | 67 | 49 |

Furthermore, we have added Figure 11 to the newly submitted PDF, which presents more visual comparisons on 3D VAE reconstruction. The results demonstrate that our method achieves better reconstruction quality than both Direct3D-S2 and TripoSF. We hope this newly added content will help the AC better understand the merits of our approach.

---

### Author Response · Authors · 2025-12-03
**Summary**

Dear PCs, SACs, ACs, and Reviewers,

We sincerely thank the reviewers for their detailed and constructive feedback. During the rebuttal period, we have comprehensively addressed all key concerns and included detailed experimental evidence to further support the core claims and contributions of our proposed method. We believe that the revised version now meets the acceptance criteria in terms of technical novelty, experimental completeness, and efficiency improvement.

The following is a summary of the two core issues of our submission, focusing on the innovation of our method, the issues we have addressed, and our counterarguments to the lower scores.

# The novelty of our method

In this paper, we propose Focusing, an efficient training framework for 3D Variational Autoencoders based on visibility. By performing visibility-based voxel carving in the latent space, our method significantly reduces unnecessary computations during the decoding and mesh extraction processes. Furthermore, by implicitly reducing the number of voxels, it indirectly enhances the network's ability to capture geometric details.

We appreciate reviewers j9yh, RLB5, and CReW for acknowledging the innovation of our method. While reviewer YxmK considered our approach to be incremental compared to TripoSF, we would like to clarify that our method is a simple and effective rendering-based acceleration module for 3D VAE training, and it is not limited to being used with TripoSF. For instance, Sparc3D also employs a rendering loss to supervise 3D VAE training. However, since Sparc3D is not open-source, we only tested our module within the TripoSF framework.


# Comparison with More Methods

All four reviewers noted that our method lacked comparison with more existing approaches. Specifically, Reviewer RLB5 considered this not to be a "major concern" and assigned a score of 8. In contrast, Reviewer j9yh argued that our paper was incomplete and that accepting it would be unfair, thus giving a score of 2.

We first need to clarify that the primary reason for the limited comparative evaluation was the difference in experimental setups and evaluation methodologies between other methods and our work. We stated in the original submission that only TripoSF and our method utilize raw mesh as input and output a result aligned with the raw mesh. **Comparing our work with other methods presents a potential issue of unfairness**. Recognizing that the reviewers may not be fully familiar with the differences between 3D VAEs, we have provided a more detailed explanation in the revision and have included the relevant comparative results in the Appendix.

Furthermore, we sincerely thank Reviewer j9yh for recognizing our idea as "*meaningful,*" "*well-motivated,*" and a "*complement to existing work,*" and for assigning positive scores for Soundness (3), Presentation (3), and Contribution (2). However, we respectfully disagree with Reviewer j9yh's sentiment that the initial submission was incomplete due to a lack of sufficient experiments, deeming acceptance unfair. We explained in the original submission that **we primarily compare with TripoSF, as currently only TripoSF supports raw mesh reconstruction**. We further specifically explained in the rebuttal why certain recent approaches were not included as baselines: Sparc3D has no publicly available implementation, Trellis and Dora-VAE are released only at 256 resolution, and Direct3D-S2 requires watertight meshes as input. Importantly, **the original experiments already validated our main technical contribution: a substantial reduction in VAE memory usage that enables high resolution 3D generation. These findings do not depend on the additional baselines included in the revision.**


**Conclusion**: We believe that the revised paper has fully addressed all key concerns raised by the reviewers and has significantly enhanced its completeness, rigor, and persuasiveness. With these comprehensive supplementary experiments and technical clarifications, we assert that the paper is now a mature and important technical contribution.

---

### Meta-Review · Area_Chair_vDTA · 2026-01-01

**Summary:**

The paper received mixed initial reviews, with scores of 8, 4, 4, and 2. Reviewers generally acknowledged the solid engineering effort and appreciated the overall quality and efficiency improvements over TripoSF. The use of view-consistent voxel carving and adaptive zooming was viewed as practically useful. At the same time, multiple reviewers raised substantive concerns about the incremental nature of the contribution, limited baseline diversity, and—most notably—the lack of clear ablation evidence isolating which design components are responsible for the observed quality improvements.

In the rebuttal, the authors provided a substantial number of additional experiments and clarifications. These include comparisons to additional baselines, expanded efficiency analyses,  sensitivity studies for carving thresholds, etc. While these additions address several concerns related to efficiency, completeness, and presentation, a key issue remains only partially resolved. In particular, although component-wise ablations were added, they primarily emphasize efficiency-related effects, and it remains unclear which component—voxel carving, adaptive zooming, loss design, or other factors—plays the dominant role in improving reconstruction quality. As a result, the AC anticipates that final reviewer opinions will likely remain mixed, with at least two reviewers maintaining their original negative scores (see detailed discussion in Reviewer Concerns and Reviewer Scores).

From the AC’s perspective, while the overall idea is reasonable and the improvements over TripoSF are notable, the submission does not yet convey a clear and well-isolated technical message about what fundamentally drives the performance gains. Without convincingly disentangling the contributions of the individual components, it is difficult to assess the true novelty and significance of the proposed method or to provide clear guidance to the community. Given this shared concern about ablation clarity and contribution attribution among multiple reviewers, the AC finds that the paper does not meet the bar for acceptance in its current form and therefore recommends rejection, while encouraging the authors to strengthen the analysis to more clearly identify and communicate the key factors responsible for the improvements.

**Reviewer Concerns:**

### Reviewer YxmK (Score: 4)

- The reviewer raised concerns about the incremental nature of the contribution relative to TripoSF, arguing that depth-based carving and adaptive zooming are primarily engineering improvements. They questioned whether the method adequately preserves or reconstructs internal structures, noted limited baseline coverage (with comparisons largely restricted to TripoSF), and felt that the quantitative gains were relatively modest. Additional concerns included missing or insufficient ablations (e.g., carving thresholds, zoom schedules, TSDF warm-up) and a lack of detailed VRAM and runtime breakdowns.
- In the rebuttal, the authors clarified that internal voxels are preserved at inference time and supported this with new visualizations. They added comparisons to additional methods where feasible, provided new ablations on carving and training components, and included detailed memory usage and training-speed analyses. They also refined several claims to better align with the empirical improvements.

---

### Reviewer j9yh (Score: 2)

- The reviewer’s main concerns centered on evaluation completeness, arguing that the original submission lacked sufficient efficiency analysis (runtime and VRAM) and ablations disentangling the effects of voxel carving, zooming, and loss terms. They also raised concerns about paper completeness and fairness, stating that adding many experiments during rebuttal was unfair, and requested clarification on multiple technical details in the formulation.
- In response, the authors added comprehensive efficiency analyses, component-wise ablations, and clarified all technical questions raised. They explicitly addressed the procedural concern by citing ICLR guidelines that encourage adding experiments during the discussion period, and revised or clarified claims and notation accordingly.

---

### Reviewer RLB5 (Score: 8)

- The reviewer requested broader baseline comparisons, clarification on gradient flow through the hard voxel carving operation, and more standardized efficiency evaluations. They also noted potential issues with thin structures and view bias.
- In the rebuttal, the authors added additional comparisons and heatmap visualizations, explained the carving operation as analogous to hard pruning or dropout with indirect gradient sharing via attention, provided standardized efficiency tables, and included discussion and examples of failure cases involving thin structures.

---

### Reviewer CReW (Score: 4)

- The reviewer raised concerns about insufficient experimental validation in the original submission, including missing ablations isolating individual components, lack of quantitative efficiency evaluation, limited generative comparisons, and absence of sensitivity analysis for carving thresholds.
- In the rebuttal, the authors added component-wise ablations (carving, zooming, and regularization), quantitative evaluations of memory usage and training speed, sensitivity analysis for carving thresholds, and expanded qualitative and quantitative comparisons.

**Reviewer Scores:**

### Reviewer YxmK

- **Original score:** 4
- **Predicted final score:** 4
- **Rationale:** The rebuttal addresses many of the reviewer’s requests by adding ablations, efficiency analyses, and clarifications regarding internal structure preservation. However, the reviewer’s request for ablations primarily concerned isolating design choices that contribute to reconstruction quality, whereas the major newly added ablations (Table 3) focus on efficiency-related factors. In addition, the reviewer’s skepticism regarding novelty and the perceived incremental nature of the contribution may persist, making a score increase unlikely.

---

### Reviewer j9yh

- **Original score:** 2
- **Predicted final score:** 2–4
- **Rationale:** The rebuttal substantially expands the experimental evaluation and attempts to address the reviewer’s technical questions. The request for broader baseline comparisons appears to be largely resolved. However, the new ablation results may not fully address the reviewer’s concern about which components (carving, zooming, or loss design) are responsible for the observed quality improvements. Moreover, the reviewer appears to hold a view regarding the incompleteness of the original submission. As a result, the score is likely to remain negative, with a possible increase to 4.

---

### Reviewer RLB5

- **Original score:** 8
- **Predicted final score:** 8
- **Rationale:** The reviewer was already strongly positive, and the rebuttal addresses the clarificatory points they raised. The score is likely to remain unchanged.

---

### Reviewer CReW

- **Original score:** 4
- **Predicted final score:** 4–6
- **Rationale:** The rebuttal directly addresses the reviewer’s requests for ablations, efficiency analysis, and sensitivity studies. However, similar to other reviewers, this reviewer requested ablations to study the quality impact of individual components, which may not be fully addressed by the added experiments. As a result, a score increase to 6 is possible but not guaranteed.

---

### Decision · Program_Chairs · 2026-01-26

Reject